# Nonparametric Identification of Latent Concepts

**Yujia Zheng** [1] [*]   **Shaoan Xie** [1] [*]   **Kun Zhang** [1] [2]

## Abstract

We are born with the ability to learn concepts by comparing diverse observations. This helps us to understand the new world in a compositional manner and facilitates extrapolation, as objects naturally consist of multiple concepts. In this work, we argue that the cognitive mechanism of comparison, fundamental to human learning, is also vital for machines to recover true concepts underlying the data. This offers correctness guarantees for the field of concept learning, which, despite its impressive empirical successes, still lacks general theoretical support. Specifically, we aim to develop a theoretical framework for the identifiability of concepts with multiple classes of observations. We show that with sufficient diversity across classes, hidden concepts can be identified without assuming specific concept types, functional relations, or parametric generative models. Interestingly, even when conditions are not globally satisfied, we can still provide alternative guarantees for as many concepts as possible based on local comparisons, thereby extending the applicability of our theory to more flexible scenarios. Moreover, the hidden structure between classes and concepts can also be identified nonparametrically. We validate our theoretical results in both synthetic and real-world settings.

## 1. Introduction

Humans possess an innate ability to learn concepts, i.e., conceptual factors underlying the observational world, by comparing diverse classes of observations, a process foundational to cognitive development (Rosch, 1973; Fodor & Pylyshyn, 1988). For example, a child distinguishes between different types of animals not by memorizing each species separately, but by observing and comparing dif-

[*]Equal contribution [1]Carnegie Mellon University [2]Mohamed bin Zayed University of Artificial Intelligence.

*Proceedings of the 42nd International Conference on Machine Learning*, Vancouver, Canada. PMLR 267, 2025. Copyright 2025 by the author(s).

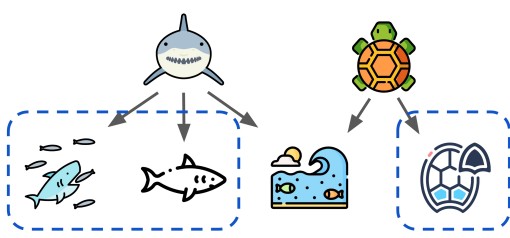

Figure 1: The class "shark 🦈" has concepts like "predator 🦈," "sleek body ⤳," and "ocean 🏞." The class "turtle 🐢" has concepts like "shell 🐚" and "ocean 🏞." A child may learn to distinguish between these classes by focusing on the unique concepts specific to each—such as "predator 🦈" and "sleek body ⤳" for "shark 🦈," and "shell 🐚" for "turtle 🐢."

ferences between various species, thereby identifying the unique concepts that define each group (e.g., Fig. 1). This mechanism of learning through comparison has been extensively studied and verified across various fields, including psychology and neuroscience, affirming its universality and effectiveness (Bruner et al., 1957).

Meanwhile, in machine learning, the extraction of conceptual features is crucial for the development of robust and interpretable models, illustrating the integration of cognitive principles into machine intelligence (Valiant, 1984; Mitchell, 1997). Recent research has achieved notable success in deriving human-interpretable concepts from various data modalities with different formulations of the problem (Bau et al., 2017; Radford et al., 2017; Alvarez Melis & Jaakkola, 2018; Kim et al., 2018; Zhou et al., 2018; Yeh et al., 2020; Koh et al., 2020; Du et al., 2021; Bai et al., 2022; Achtibat et al., 2022; Crabbé & van der Schaar, 2022; Liu et al., 2023; Park et al., 2023; Jiang et al., 2024). These concepts have proven beneficial for tasks such as extrapolation (Janner et al., 2022; Lachapelle et al., 2023; Du & Kaelbling, 2024), explanation (Alvarez Melis & Jaakkola, 2018; Sreedharan et al., 2020; Leemann et al., 2023; Poeta et al., 2023), and decision-making (Grupen et al., 2022; Zabounidis et al., 2023; Delfosse et al., 2024). Furthermore, advancements in this domain have significantly contributed to scientific discovery, particularly in healthcare (Jia et al., 2022).

While numerous methods have been developed to extract concepts from data, most provide only empirical support and lack theoretical guarantees concerning the correctness

of the recovered concepts (Marconato et al., 2024b). With the help of specific parametric assumptions, a few studies have explored the identifiability of concept learning. For example, with the linear representation hypothesis that concepts are linearly related, recent research (Rajendran et al., 2024; Reizinger et al., 2024; Marconato et al., 2024a) has shown that the concept space can be identified up to a linear transformation. Another line of research has tackled object-centric learning, attempting to identify individual objects as groups of pixels (slots), such as trees or dogs, while excluding more abstract concepts like lighting and styles. In addition to these concept type restrictions, further assumptions are also required for identifiability, such as no occlusion between objects (each observed variable is influenced by only one concept) (Brady et al., 2023; Wiedemer et al., 2024) or the additivity of the generating process (Lachapelle et al., 2023; Wiedemer et al., 2024). These studies mark significant exploration toward understanding concept learning. At the same time, the constraints imposed on concept types and functional relationships may limit the confidence to fully account for the empirical success observed in concept learning from real-world scenarios. Therefore, despite significant empirical progress, a fundamental question in concept learning remains unanswered:

*In the general setting, which concepts can we reliably recover with theoretical guarantees?*

We address this question by grounding our approach in a fundamental cognitive mechanism: humans learn concepts by contrasting diverse classes of observations. Classic studies have shown that concept formation relies on detecting distinctions across examples: Bruner et al. (1957) emphasized learning through contrasts between exemplars and non-exemplars; Gibson (Gibson, 1963; 1969) proposed differentiation as a basis of perceptual learning in infants; and Gentner & Namy (1999) demonstrated that direct comparison enables children to abstract category-defining features. This principle is further supported by extensive literature across cognitive science, reinforcing that it is only through discerning the differences between classes that humans can unravel and understand previously unseen concepts. As a result, in the most general setting, the essential information for provably learning hidden concepts must pertain to the diversity present among different classes.

Inspired by this cognitive process of learning by comparison, we establish a set of theoretical guarantees on concept learning in the general setting. We show that hidden concepts can be identified without relying on assumptions about the nature of the concepts or specific parametric models, provided there is sufficient diversity across classes. Specifically, we first prove that for any pair of classes, the unique part of the concepts for each class can be disentangled from the remaining concepts (Thm. 1). This pairwise comparison[1] serves as a foundational prototype for learning concepts, enabling the flexible identifiability of as many concepts as possible, given that they exhibit enough diversity, even when others do not. We then extend the pair-wise identifiability to learn unique concepts from an arbitrary subset of classes (Cor. 1). Given that most works rely on global assumptions for all concepts and fail to offer guarantees when assumptions are partially violated for some concepts, the proposed flexible identifiability by local comparisons provides unique practical value, since real-world scenarios often do not perfectly conform to ideal conditions for all concepts.

Furthermore, with sufficient diversity across different classes of observations, we prove the nonparametric identifiability for all class-related hidden concepts up to an element-wise transformation and permutation (Thm. 2). For other invariant background concepts, such as "chromatic" that remain consistent across all classes, we can also identify them under appropriate structural diversity conditions (Prop. 1). Consequently, we introduce, to the best of our knowledge, one of the first frameworks for concept identifiability in the general setting that does not confine itself to specific concept types or parametric generative models. Moreover, the connective structure between classes and concepts can also be recovered in a nonparametric way (Prop. 2). Our theoretical results are substantiated through empirical validation on synthetic data and five different real-world datasets.

## 2. Preliminaries

We first introduce the problem setting as well as some essential notation. Fig. 2 illustrates the key notation and relations of the considered setting. We also provide a structured summary of notation in Appx. A for a quick reference.

**Data-generating process.** Let $\mathbf{x} = (\mathbf{x}_1, \ldots, \mathbf{x}_m) \in \mathcal{X} \subseteq \mathbb{R}^m$ be a vector representing observed variables. We assume that the observation $\mathbf{x}$ is generated by latent *concept* variables $\mathbf{z} = (\mathbf{z}_A, \mathbf{z}_B) \in \mathcal{Z} \subseteq \mathbb{R}^n$. The generating process is as follows, with Fig. 2 serving as a concrete example:

$$\mathbf{x} := f(\mathbf{z}), \tag{1}$$

where $\mathbf{z}$ consists of class-dependent part $\mathbf{z}_A = (\mathbf{z}_1, \ldots, \mathbf{z}_{n_A}) \in \mathcal{Z}_A \subseteq \mathbb{R}^{n_A}$ and class-independent part $\mathbf{z}_B = (\mathbf{z}_{n_A+1}, \ldots, \mathbf{z}_n) \in \mathcal{Z}_B \subseteq \mathbb{R}^{n_B}$, $n_B = n - n_A$. These two parts are conditionally independent given observed *class* variables $\mathbf{c} = (\mathbf{c}_1, \ldots, \mathbf{c}_u) \subseteq \mathbb{R}^u$, i.e., $p(\mathbf{z}|\mathbf{c}) = p(\mathbf{z}_A|\mathbf{c})p(\mathbf{z}_B)$. Since $\mathbf{z}_A$ depends on the classes $\mathbf{c}$, we represent $\mathbf{z}_A := g(\mathbf{c}, \theta)$, where $\theta$ denotes other factors including potential noise. All densities are smooth and positive, and domains are path-connected.

---

[1]Learning by comparison serves as an inspiration for identifiability theory, rather than being a specific estimation method.

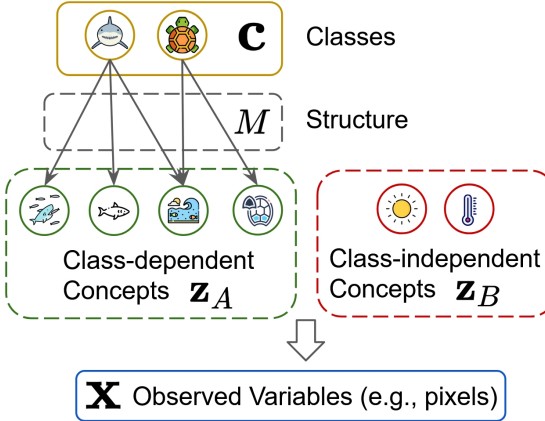

Figure 2: The problem setting. Consider images of aquatic animals, where the observed variables $\mathbf{x}$ represent image pixels. The different animal types (e.g., "shark 🦈" and "turtle 🐢") correspond to class variables $\mathbf{c}$. Class-dependent concept variables $\mathbf{z}_A$ might include attributes like "predator 🦈," "sleek body ➪," "ocean 🏞" and "shell 🐢" (see, e.g., Fig. 1), while class-independent concept variables $\mathbf{z}_B$ could be "lighting ☀" and "temperature 🌡." The hidden generative process of each image depends on all of these concepts, though only some are specific to each class (encoded by the structure $M$, which is a binary adjacency matrix). The goal is to identify $\mathbf{z}$ based on observed variables $\mathbf{x}$ and classes $\mathbf{c}$.

The generating function $f$ is an *unknown* diffeomorphism onto its image, capturing complex mixing in observational data without imposing specific distributional assumptions, such as Gaussian, on latent concepts $\mathbf{z}$. This allows for a broad formulation that encompasses diverse concepts and nonparametric generative models. Equation 1 can also be extended to the additive noise setting through a standard deconvolution technique (Khemakhem et al., 2020a).

**Technical notation.** Throughout this work, for any matrix $S$, we use $S_{i,\cdot}$ for its $i$-th row and $S_{\cdot,j}$ for its $j$-th column. We denote the first $k$ dimensions of the $i$-th row as $S_{i,\cdot k}$ and the remaining as $S_{i,k+1\cdot}$. Similarly, the first $k$ dimensions of the $j$-th column are $S_{\cdot k,j}$ and the remaining $S_{k+1\cdot,j}$. For any set of indices $\mathcal{I} \subset \{1,\ldots,m\} \times \{1,\ldots,n\}$, analogously, we have $\mathcal{I}_{i,\cdot} := \{j \mid (i,j) \in \mathcal{I}\}$ and $\mathcal{I}_{\cdot,j} := \{i \mid (i,j) \in \mathcal{I}\}$. We define the support of $S \in \mathbb{R}^{a \times b}$ as $\text{supp}(S) := \{(i,j) \mid S_{i,j} \neq 0\}$, and also extend $\text{supp}(\cdot)$ to a matrix-valued function $\mathbf{S}(\cdot)$, defining $\text{supp}(\mathbf{S}) := \{(i,j) \mid \exists \theta \in \Theta, \mathbf{S}(\theta)_{i,j} \neq 0\}$. Then we define $\mathcal{D}$ as the support of $D_{\mathbf{c}}g$, where $D_{\mathbf{c}}g$ represents the partial derivative of $g$ w.r.t. $\mathbf{c}$.

*Example* 1. Consider the matrices

$$S = \begin{bmatrix} 1 & 0 & 3 \\ 0 & 0 & 5 \\ 4 & 0 & 0 \end{bmatrix}, \quad \mathbf{S}(\theta) = \begin{bmatrix} \theta & 0 & 3 \\ 0 & 0 & 5\theta \\ 4 & 0 & 0 \end{bmatrix}.$$

The support of the matrix $S$ is $\text{supp}(S) = \{(1,1),(1,3),(2,3),(3,1)\}$. For the matrix-valued function $\mathbf{S}(\theta)$, the support $\text{supp}(\mathbf{S})$ remains the same as long as some $\theta \neq 0$ makes those entries nonzero.

**Connective structure.** Based on these, we define the *structure* $M$ as a binary matrix with the support $\mathcal{D}_{\cdot n_A,\cdot}$. The class-dependent part $\mathbf{z}_A$ can be further represented as

$$p(\mathbf{z}_A|\mathbf{c}) = \prod_{i=1}^{n_A} p(\mathbf{z}_i | M_{i,\cdot} \odot \mathbf{c}), \tag{2}$$

where $M_{i,\cdot}$ is the $i$-th row of $M$. The operator $\odot$ denotes the element-wise (Hadamard) product. Since classes $\mathbf{c}$ are not connected to class-independent part $\mathbf{z}_B$, $M$ illustrates the connective structure between classes $\mathbf{c}$ and concepts $\mathbf{z}$. The conditional independence provides a form of modularity commonly adopted in prior work on identifiable latent variable models (Hyvärinen & Morioka, 2016; Khemakhem et al., 2020a; Sorrenson et al., 2020; Lachapelle et al., 2022; Hyvärinen et al., 2024). It may be particularly natural in our class-concept framework; for example, while the concepts "wings" and "feathers" are related, they become conditionally independent given the class variable "bird." Moreover, we define $A_i$ as the index set of concepts corresponding to class $\mathbf{c}_i$, with the associated concepts represented as $\mathbf{z}_{A_i}$. Likewise, $\mathbf{z}_{A_i \setminus A_j}$ refers to the difference in the concept sets between classes $\mathbf{c}_i$ and $\mathbf{c}_j$.

*Example* 2. Consider the example in Fig. 2, we have $\mathbf{z}_A = (\mathbf{z}_1 \text{🦈}, \mathbf{z}_2 \text{➪}, \mathbf{z}_3 \text{🏞}, \mathbf{z}_4 \text{🐢})$ and classes $\mathbf{c} = (\mathbf{c}_1 \text{🦈}, \mathbf{c}_2 \text{🐢})$. Then we have $\mathbf{z}_{A_1} = \{\mathbf{z}_1, \mathbf{z}_2, \mathbf{z}_3\}$ and $\mathbf{z}_{A_2} = \{\mathbf{z}_3, \mathbf{z}_4\}$, and $\mathbf{z}_{A_1 \setminus A_2} = \{\mathbf{z}_1, \mathbf{z}_2\}$. The connective structure $\mathbf{M}$ is a binary matrix:

$$\mathbf{M} = \begin{bmatrix} 1 & 1 & 1 & 0 \\ 0 & 0 & 1 & 1 \end{bmatrix}^\top.$$

**Observational equivalence.** All of our identifiability results are rooted in the observational equivalence between the ground truth and the estimated model. This equivalence can be achieved in the large-sample limit using maximum likelihood estimation, which is *estimator-agnostic* and can be facilitated by methods such as normalizing flows.

**Definition 1** (Observational Equivalence). Two models $(g, p_{\mathbf{z}}, M)$ and $(\hat{g}, p_{\hat{\mathbf{z}}}, \hat{M})$ are observationally equivalent if and only if $p_{\hat{\mathbf{x}}|\mathbf{c}}(x|c) = p_{\mathbf{x}|\mathbf{c}}(x|c)$.

## 3. Identifiability Theory

Without any assumptions on specific concept types, functional relations, or parametric generative models, to what extent can we provably learn hidden concepts from diverse classes of observations?

To answer this, in Sec. 3.1, we first prove that the unique concepts in any pair of classes can be disentangled from

the remaining ones (Thm. 1). Based on this, we can fully leverage the diversity in the data and provide flexible identifiability for any subset of concepts, as long as there exists sufficient diversity for local comparison (Cor. 1). For the global identification, in Sec. 3.2, we prove the nonparametric identifiability for all class-dependent hidden concepts (Thm. 2) under the structural diversity condition (Assump. 1). Together with a sparsity condition for the remaining class-independent part, all hidden concepts can be identified up to trivial indeterminacy (Prop. 1). Furthermore, in Sec. 3.3, we recover the hidden connective structure between classes and concepts (Prop. 2), providing further insights into the latent compositional relations.

To enhance technical understanding, we provide a detailed discussion in each section on connections to techniques from the broader identifiability literature. Additional concrete examples and discussions are available in Appx. D.

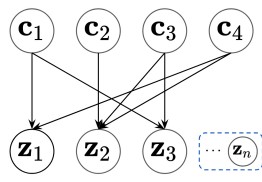

Figure 3: Structure in Running Example 3.

*Example* 3. We introduce a running example (Fig. 3) to illustrate key implications and insights throughout the paper. The lines depict the connective structure between classes and concepts, distinguishing class-dependent concepts, $\mathbf{z}_A = (\mathbf{z}_1, \mathbf{z}_2, \mathbf{z}_3)$, from class-independent variables, $\mathbf{z}_B$, enclosed within the blue dotted square.

### 3.1. Learning Concepts by Local Comparison

Humans learn concepts by leveraging the diversity across classes. We argue that the fundamental mechanism in this cognitive process is learning through pair-wise comparison, since any two classes can only be distinguished by identifying their unique concepts. Pairwise comparison thus serves as the basic unit for concept learning across multiple classes, as comparisons among any set of classes can be reduced to pairs. In the following theorem, we prove that the unique concepts between any pair of classes can be disentangled from the remaining concepts. The proof is in Appx. B.1.

We first introduce some additional notation. Let us define T as a matrix with the same support of $\mathbf{T}(\cdot)$ in $D_{\mathbf{c}}\hat{g} = \mathbf{T}D_{\mathbf{c}}g$, where $\mathbf{T}(\cdot)$ is a matrix-valued function. In addition, given a subset $\mathcal{S} \subseteq \{1, \ldots, n\}$, the subspace $\mathbb{R}^n_{\mathcal{S}}$ is defined as:

$$\mathbb{R}^n_{\mathcal{S}} := \{s \in \mathbb{R}^n \mid s_i = 0 \text{ if } i \notin \mathcal{S}\}, \tag{3}$$

where $s_i$ is the $i$-th element of the vector $s$.

*Example* 4. Intuitively, $\mathbb{R}^n_{\mathcal{S}}$ consists of all vectors in $\mathbb{R}^n$ where only the coordinates indexed by $\mathcal{S}$ can vary, while the remaining coordinates are fixed at zero. For Example 3, if $n = n_A = 3$ and $\mathcal{S} = \hat{\mathcal{D}}_{\cdot:i} = \{1,3\}$, then $\mathbb{R}^{n_A}_{\hat{\mathcal{D}}_{\cdot:i}} = \mathbb{R}^3_{\{1,3\}}$

consists of vectors of the form $(s_1, 0, s_3)$, where $s_1, s_3 \in \mathbb{R}$ are free to take any real values.

**Theorem 1** (Learning by pairwise comparison). *Consider two **observationally equivalent** (Defn. 1) models $(g, p_{\mathbf{z}}, M)$ and $(\hat{g}, p_{\hat{\mathbf{z}}}, \hat{M})$ as in Sec. 2. Suppose, with an $\ell_0$ regularization on $D_{\mathbf{c}}\hat{g}$ ($|\hat{\mathcal{D}}| \leq |\mathcal{D}|$), there exist a set of points $\{(c, \theta)^{(\ell)}\}_{\ell=1}^{|\mathcal{D}_{\cdot,i}|}$, such that*

1. $\{D_{\mathbf{c}}g((c, \theta)^{(\ell)})_{\cdot,i}\}_{\ell=1}^{|\mathcal{D}_{\cdot,i}|}$, *are linearly independent;*

2. $\left[\mathrm{T}D_{\mathbf{c}}g((c, \theta)^{(\ell)})\right]_{\cdot,i} \in \mathbb{R}^{n_A}_{\hat{\mathcal{D}}_{\cdot,i}}$.

*Then for any pair of classes $\mathbf{c}_i$ and $\mathbf{c}_j$, there exists a permutation $\pi$ such that the unique concepts of each class are disentangled with other concepts, i.e., $\frac{\partial \hat{\mathbf{z}}_{\pi(A_i \setminus A_j)}}{\partial \mathbf{z}_{A_j}}$ and $\frac{\partial \hat{\mathbf{z}}_{\pi(A_j \setminus A_i)}}{\partial \mathbf{z}_{A_i}}$ equal to zero.*

**Insights.** For Example 3, given classes $\mathbf{c}_1$ and $\mathbf{c}_3$, Thm. 1 ensures that for any pair of classes, such as $\mathbf{c}_1$ and $\mathbf{c}_2$, the concept unique to $\mathbf{c}_1$, $\mathbf{z}_1$, can be disentangled from the other concepts $\{\mathbf{z}_2, \mathbf{z}_3\}$. Similarly, the concept unique to $\mathbf{c}_3$, $\mathbf{z}_2$, can be disentangled from the other concepts $\{\mathbf{z}_1, \mathbf{z}_3\}$, up to a standard permutation indeterminacy.

We have also extended the guarantees for pairwise comparisons to arbitrary class sets, facilitating more efficient learning in complex scenarios with its proof in Appx. B.2

**Corollary 1** (Learning by local comparison). *Consider two **observationally equivalent** (Defn. 1) models $(g, p_{\mathbf{z}}, M)$ and $(\hat{g}, p_{\hat{\mathbf{z}}}, \hat{M})$ as in Sec. 2. Suppose that the assumptions in Thm. 1 hold. Then, for a set of classes $\mathbf{c}_I$ and its corresponding concept sets $\mathbf{z}_{A_I}$ with a set of indices $I$, there exists a permutation $\pi$ such that the unique concepts for the class $\mathbf{c}_i$ are disentangled with concepts associated with other classes, i.e., $\frac{\partial \hat{\mathbf{z}}_{\pi(A_i \setminus A_{I \setminus i})}}{\mathbf{z}_{A_{I \setminus i}}} = 0$.*

**Insights.** For Example 3, given a set of classes $\mathbf{c}_I = \{\mathbf{c}_2, \mathbf{c}_3, \mathbf{c}_4\}$, Cor. 1 ensures that for any class in the set, e.g., $\mathbf{c}_4 \in \mathbf{c}_I$, the unique concept $\mathbf{z}_1$ can be disentangled from the concepts associated with others, i.e., $\{\mathbf{z}_2, \mathbf{z}_3\}$, where $\mathbf{c}_I$ can be a arbitrary subset of all classes.

**Discussion on nondegenerate sample space.** A key imprint of the nonparametric generating process in observational data is its structure. The assumption helps ensure the possibility of capturing the connective structure by the Jacobian in the nonlinear cases, following the similar spirit in (Lachapelle et al., 2022; Zheng et al., 2022). In general, it avoids pathological cases where all samples originate from highly restricted sub-populations that only cover a degenerate subspace. The first part makes sure that there are at least $|\mathcal{D}_{\cdot n_A, i}|$ data points such that the Jacobian function spans the

support space, which is almost always guaranteed asymptotically. The second part $\left[ \mathbf{T} D_\mathbf{c} g((\mathbf{c}, \theta)^{(\ell)}) \right]_{\cdot, i} \in \mathbb{R}^{n_A}_{\hat{\mathcal{D}}_{\cdot, i}}$ is also mild since $\hat{\mathcal{D}}_{\cdot, i} = \mathbf{T} D_\mathbf{c} g((\mathbf{c}, \theta)^{(\ell)})$ always resides in $\mathbb{R}^{n_A}_{\hat{\mathcal{D}}_{\cdot, i}}$. Even in some rare cases where the matrix does not fit the support due to some generic combination of values, the assumption is still almost always satisfied asymptotically, since it only necessitates the existence of one matrix in the entire space with the same support of $\mathbf{T}(\cdot)$. Meanwhile, the non-degeneracy assumption excludes linear transformations with constant Jacobians, as linear Gaussian models are non-identifiable and linear non-Gaussian models have already been extensively studied. An illustrative example is provided as Example 5 in Appx. D.2.

**Partial identifiability benefits applicability.** Besides being the foundation for the learning process, the principles of local comparisons in both Thm. 1 and Cor. 1 also enable partial identifiability for a subset of concepts when diversity is not universally satisfied for all. Previous theoretical studies on concept learning often assume universal conditions like linearity or additivity. Nevertheless, such assumptions rarely hold for all in complex, unpredictable real-world scenarios and fail to guarantee identifiability under any degree of violation. This challenge of handling partial assumption violations persists in the broader identifiability literature (Hyvärinen & Morioka, 2016; Khemakhem et al., 2020a; Taeb et al., 2022; Zheng et al., 2022; Hyvärinen et al., 2024; Zhang et al., 2024; Zheng et al., 2025). Fortunately, with the proposed theory based on local comparisons (Thm. 1 and Cor. 1), we can leverage the structure to recover the hidden system as much as possible, even when the degree of diversity does not support global identifiability. For instance, while shared concepts across similar classes remain unidentifiable, sufficient diversity ensures identifiability for other concepts. Crucially, these new flexible guarantees require no additional restrictive assumptions on concept types, functional forms, or parametric models.

### 3.2. Learning Concepts by Global Comparison

Inspired by the mechanism of local comparison, we have shown that it is possible to fully leverage the diversity among different classes of observations to recover hidden concepts as much as possible. This naturally leads us to consider the conditions required for identifying all hidden concepts in a global manner. We first prove that, under the condition of *Structural Diversity* (Assump. 1), all class-dependent concepts are identifiable up to a composition of a permutation and an element-wise invertible transformation (Thm. 2). The proof is included in Appx. B.3.

**Assumption 1.** (Structural Diversity) For any $\mathbf{z}_i \in \mathbf{z}_A$, there exists a set of indices $J$ ($|J| > 1$) and $j \in J$ where $M_{i,j} \neq 0$ and $M_{i,k} = 0$ for all $k \in J$, $k \neq j$, and $M_{\cdot, J \setminus \{j\}}$ is the only row with all zero entries in $M_{\cdot, J \setminus \{j\}}$.

**Theorem 2** (Learning by global comparison). *Consider two models $(g, p_\mathbf{z}, M)$ and $(\hat{g}, p_{\hat{\mathbf{z}}}, \hat{M})$ as in Sec. 2. Under the assumptions in Thm. 1 and Assump. 1, suppose that for any set $A_\mathbf{z} \subseteq \mathcal{Z}$ with non-zero probability measure and which cannot be expressed as $B_{\mathbf{z}_B} \times \mathbf{z}_A$ for any $B_{\mathbf{z}_B} \subseteq \mathcal{Z}_B$, there exist two values of $\mathbf{c}$, i.e., $c^{(k)}$ and $c^{(v)}$ (which may vary across different $A_\mathbf{z}$), that*

$$\int_{\mathbf{z} \in A_\mathbf{z}} p(\mathbf{z} \mid c^{(k)}) d\mathbf{z} \neq \int_{\mathbf{z} \in A_\mathbf{z}} p(\mathbf{z} \mid c^{(v)}) d\mathbf{z}.$$

*If $(g, p_\mathbf{z}, M)$ and $(\hat{g}, p_{\hat{\mathbf{z}}}, \hat{M})$ are **observationally equivalent** (Defn. 1), then:*

1. *(Element-wise Identifiability) For any $\mathbf{z}_i \in \mathbf{z}_A$, there exists a permutation $\pi$ and invertible functions $h_i : \mathbb{R} \to \mathbb{R}$ s.t. $\hat{\mathbf{z}}_i = h_i(\mathbf{z}_{\pi(i)})$;*

2. *(Block-wise Identifiability) For $\mathbf{z}_B$, there exists an invertible function $h : \mathbb{R}^{n_B} \to \mathbb{R}^{n_B}$ s.t. $\hat{\mathbf{z}}_B = h(\mathbf{z}_B)$.*

**Insights.** For Example 3, element-wise identifiability ensures all class-dependent concepts can be identified up to permutation, i.e., concept $\mathbf{z}_1$, $\mathbf{z}_2$, and $\mathbf{z}_3$ can be individually disentangled from each other. The block-wise indeterminacy makes sure that the class-independent concepts $\mathbf{z}_B = (\mathbf{z}_4, \ldots, \mathbf{z}_n)$ as a block (group of variables) can be disentangled from the class-dependent concepts $\mathbf{z}_A$. These two types of identifiabilities are standard in the literature (Hyvärinen & Morioka, 2017; Lachapelle et al., 2022; Von Kügelgen et al., 2021; Zheng & Zhang, 2023).

**Discussion on structural diversity.** Assumption 1, termed *Structural Diversity*, ensures sufficient diversity across different classes of observations for the nonparametric identifiability of all class-dependent concepts. Without imposing parametric constraints on concept types, functional relations, or generative models, the only exploitable information is the inherent connective structure between classes and concepts. As discussed earlier, when classes lack diversity on their corresponding concepts, identifying individual class-dependent concepts becomes impossible without extra knowledge. Thus, *Structural Diversity* is crucial for guaranteeing correctness across all concepts without relying on parametric assumptions or auxiliary information.

**Insights.** Structural diversity suggests that for each class-dependent concept $\mathbf{z}_i$, there exists a set of classes such that $\mathbf{z}_i$ is unique to one of these classes. For Example 3, the corresponding matrix $M$ is as Fig. 4. Consider $i = 1$ ($\mathbf{z}_1$), there exists a set of class indices $J = \{1, 3\}$ s.t. $M_{1,1} \neq 0$ and $M_{1,3} = 0$. Meanwhile, $M_{i, J \setminus \{j\}} = M_{1,3}$ is the only row with all zero entries in $M_{\cdot, J \setminus \{j\}} = M_{\cdot, 3}$. Thus, the structural diversity holds for concept $\mathbf{z}_1$.

The structural difference in the example above implies that $\mathbf{z}_1$ can be distinguished by considering these classes. Simultaneously, we have sufficient information for all the remaining concepts, as the submatrix $M_{.,J\setminus 1}$ encompasses the other concepts. Then it is possible

$$
\begin{array}{c c c c c}
 & \mathbf{c}_1 & \mathbf{c}_2 & \mathbf{c}_3 & \mathbf{c}_4 \\
\mathbf{z}_1 & 1 & 0 & 0 & 1 \\
\mathbf{z}_2 & 0 & 1 & 1 & 1 \\
\mathbf{z}_3 & 1 & 0 & 1 & 0
\end{array}
$$

Figure 4: Example of *Structural Diversity*.

to uniquely identify $\mathbf{z}_1$ among all the class-dependent hidden concepts. Coupled with this sufficient diversity for other concepts, we have the *Structural Diversity* assumption for the nonparametric identifiability of all class-dependent hidden concepts in a structural view.

*Comparison with pervious conditions.* Different from various assumptions encouraging the sparsity of the structure in the identifiability literature (Rhodes & Lee, 2021; Moran et al., 2021; Zheng et al., 2022; Zheng & Zhang, 2023), our assumption only ensures necessary variability on the dependency structure and could also hold true with relatively dense connections. At the same time, we permit arbitrary structures between the class-dependent hidden concepts and the observed variables, while previous work has to assume a sparse structure on the generating process between latent and observed variables. This flexibility accommodates a general generative process, thereby distinguishing our assumptions from others. Additionally, another standard identification strategy requires $2n_A + 1$ distinct domains or classes to achieve latent variable identifiability (e.g., (Hyvärinen & Morioka, 2017; Khemakhem et al., 2020a; Kong et al., 2022)), a condition we do not impose.

*Limitations.* Of course, since we aim for the general nonparametric identifiability for all class-dependent concepts, there are scenarios where it is impossible to fully recover every hidden concept, even with the help of the *Structural Diversity* condition. For example, if all dog breeds are defined by overlapping features like "barks" and "furry," a learner could not distinguish breeds without unique distinguishing traits. Here, the lack of unique concepts violates *Structural Diversity*, preventing identifiability from observational data alone. In such scenarios, prior assumptions from provable concept learning—like disjoint concept representations (e.g., non-overlapping Jacobians), linearity, or additive generating functions—remain essential to resolve ambiguities (Brady et al., 2023; Lachapelle et al., 2023; Wiedemer et al., 2024). Therefore, our assumption does *not* supersede the previous ones; rather, it offers a new direction that can be helpful for learning hidden concepts with minimal prior knowledge about the system.

**Discussion on distributional variability.** The other assumption introduced in Thm. 2 requires the existence of distributional variability for the block-wise identifiabilty of

$\mathbf{z}_B$. It necessitates at least two classes with differing conditional distributions. As discussed and empirically verified in (Kong et al., 2022), the likelihood of *all* classes having identical probability measures is exceedingly slim. Interestingly, these two classes may also vary across different $A_{\mathbf{z}}$. Therefore, this assumption is highly likely to be satisfied in real-world scenarios, as it is virtually impossible for the measures corresponding to *all* classes (e.g., all kinds of animals in a zoo) to be almost identical. A concrete example (Example 6) is provided in Appx. D.2. Furthermore, prior proof techniques typically require at least $n_A + 1$ domains or classes to establish block-wise identifiability (Zheng & Zhang, 2023; Li et al., 2024), and Kong et al. (2022) require $2n_A + 1$. In contrast, *two* domains often suffice in Thm. 2, an interesting technique that can also benefit other tasks.

**Generalized concept learning and beyond.** Extending the results on a subset of concepts (Thm. 1 and Cor. 1), Thm. 2 provides guarantees for learning all class-dependent hidden concepts. Unlike prior work restricted to parametric constraints (e.g., disjointness, linearity), our framework relies on *Structural Diversity* between classes and concepts, enabling broader applicability with sufficient diversity. This aligns with cognitive processes of learning by comparison, ensuring nonparametric recovery of latent structures. Beyond hidden concept learning, our theory also offers insights into latent variable models without prior knowledge, as it builds solely on their basic generative structure. Consequently, some findings may interest disentanglement (Hyvärinen et al., 2024), causal representation learning (Schölkopf et al., 2021), object-centric learning (Mansouri et al., 2024), and generalization (Du & Kaelbling, 2024).

**Identifying class-independent concepts.** After identifying class-dependent concepts, one may still be interested in how to provably uncover the remaining class-independent concepts, even though they may not stand out in the cognitive process due to their invariance. To address this, we present the following *informal* result with its formal version and proof in Appx. B.4, which identifies all concepts—both class-dependent and class-independent—in a nonparametric manner.

**Proposition 1** (Learning class-independent concepts; **Informal**). *Consider two **observationally equivalent** (Defn. 1) models $(g, p_{\mathbf{z}}, M)$ and $(\hat{g}, p_{\hat{\mathbf{z}}}, \hat{M})$ as in Sec. 2. Under the assumptions in Thm. 2, further assume structural conditions on the connective structure between $\mathbf{z}_B$ and $\mathbf{x}$ and an $\ell_0$ regularization. Then for any $\mathbf{z}_i \in \mathbf{z}$, there exists a permutation $\pi$ and invertible functions $h_i : \mathbb{R} \to \mathbb{R}$ s.t. $\hat{\mathbf{z}}_i = h_i(\mathbf{z}_{\pi(i)})$.*

To avoid introducing parametric assumptions, we still mainly rely on conditions on the connective structures. Since classes $\mathbf{c}$ are not connected to those class-independent concepts $\mathbf{z}_B$, the proposed structural condition on $M$ does not help identify $\mathbf{z}_B$. Thus, we leverage the structural condi-

tion between these concepts and the observed variables, as proposed in (Zheng et al., 2022). Consequently, if needed, we can provide nonparametric guarantees under appropriate structural conditions for both $\mathbf{z}_A$ and $\mathbf{z}_B$ in general settings.

### 3.3. Learning Structure Between Classes and Concepts

Furthermore, we show that the hidden structure $M$, which encodes the dependency relations between classes and concepts, can also be identified based on multiple classes of observations (Prop. 2). This process parallels human learning, where distinguishing between classes involves recovering underlying structures, such as aligning concepts with their corresponding classes. While hidden structure identification in complex systems remains an open challenge (Spirtes et al., 2000), our findings provide potential insights toward its resolution. The proof is included in Appx. B.5.

**Proposition 2** (Learning class-concept structure). *Consider two **observationally equivalent** (Defn. 1) models $(g, p_\mathbf{z}, M)$ and $(\hat{g}, p_{\hat{\mathbf{z}}}, \hat{M})$ as in Sec. 2. Suppose all assumptions in Thm. 1 hold, except Assump. 1. Then $\hat{M} = PM$ for a permutation matrix $P$.*

> **Insights.** In Example 3, the theory from previous sections ensures the identifiability of all latent concepts. With Prop. 2, we can now provably identify the connective structure $M$ between classes and concepts remains unknown, aligning the identified concepts with the corresponding classes.

All assumptions have been discussed in the previous sections. Compared to the previous theories on the identifiability of latent concepts, the recovery of the hidden connective structure does not necessitate the structural diversity assumption (Assump. 1). This allows us to uncover the structure in even more general scenarios, if the identification of latent concepts might not be of particular interest.

**Connection with structure learning.** In addition to revealing the hidden class-concept structure, if we consider the class variables $\mathbf{c}$ as exogenous to the system and the underlying concept variables $\mathbf{z}$ as general hidden variables, the dependency structure between exogenous noises and hidden variables encodes most of the structural information in the system, even if dependencies exist among hidden variables (e.g., a hidden directed acyclic graph (DAG)). In structure learning, similar strategies have been applied to recover the DAG among observed variables by first recovering the structure of how exogenous noises influence the system in both linear (Shimizu et al., 2006) and nonlinear (Reizinger et al., 2022) cases—the DAG constraint ensures the correspondence between the Jacobian of the mixing function (Zheng et al., 2022) and the adjacency matrix. It is worth noting that identifying the hidden structure in a general nonlinear system from purely observational data (i.e., without interventions) is a challenging problem that

has been open for decades (Spirtes et al., 2000; Zheng et al., 2023; 2024b;a). Although this is not the focus of our work, the insights provided here are of independent interest to researchers exploring this longstanding challenge.

## 4. Experiments

In order to show the identification of hidden concepts based on the proposed nonparametric identifiability theory, we conduct experiments on both synthetic and real-world datasets. It is noteworthy that an extensive body of research has empirically verified the ability to learn hidden concepts from various data modalities (Bau et al., 2017; Radford et al., 2017; Alvarez Melis & Jaakkola, 2018; Kim et al., 2018; Yeh et al., 2020; Koh et al., 2020; Bai et al., 2022; Achtibat et al., 2022; Crabbé & van der Schaar, 2022; Liu et al., 2023). Furthermore, the application range of concept learning is expanding significantly with recent advancements in foundation models (Park et al., 2023; Rajendran et al., 2024; Jiang et al., 2024). Our results complement previous empirical findings by verifying the proposed theory, and we refer to the extensive previous research outlined above for more applications of concept learning across various scenarios.

**Setup.** In the considered setting, different samples may correspond to different classes selected by a mask. We structure the dataset as $\{(\mathbf{x}, \mathbf{c})^{(i)}\}_{i=1}^N$, where $N$ denotes the sample size, and $\mathbf{c}^{(i)}$ is a multi-hot vector representing the classes for the data point $\mathbf{x}^{(i)}$. A mask $\mathcal{M}_{i,:} \odot \mathbf{c}^{(i)}$ is applied to account for the specific class for each sample. We employ a regularized maximum-likelihood method during estimation, following the standard approach in (Sorrenson et al., 2020). The objective function is defined as $\mathcal{L}(\theta) = \mathbb{E}_{(\mathbf{x},\mathbf{c})}[-\log p_{\hat{f}^{-1}}(\mathbf{x} \mid \mathcal{M}_{i,:} \odot \mathbf{c}) + \lambda \mathbf{R}]$, where $\lambda$ is the regularization parameter, and $\mathbf{R}$ represents the $\ell_1$ norm applied to $\hat{M}$ and, if estimating class-independent concepts, also to $D_{\hat{\mathbf{z}}}\hat{f}$. Following previous work, we use Mean Correlation Coefficient (MCC) to measure the alignment between the ground-truth and the recovered latent concepts. The results are from 10 random trials. Additional details and results are provided in Appx. C, such as identification with general noises and supplementary evaluation metrics across various settings.

**Synthetic datasets.** We conduct experiments on various synthetic datasets to verify the proposed identifiability theory. Specifically, we focus on two settings: learning all class-dependent concepts (Fig. 5) and learning all concepts including class-independent ones (Fig. 6), under appropriate conditions. For *Ours*, the observations are generated according to the assumptions required for the theory; while for *Base*, no structural conditions on either $\mathcal{M}$ or $\mathcal{F}$ have been imposed, and the other settings stay the same. Comparison with other baselines and more details are included in Appx. C. To measure the element-wise identifiability, we use the

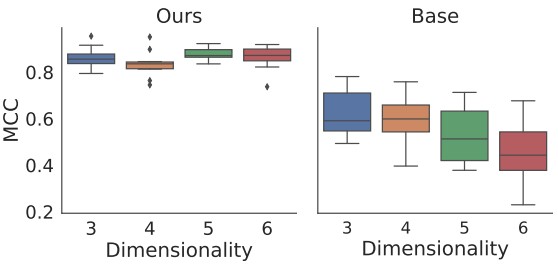

Figure 5: Identification of class-dependent concepts w.r.t. different numbers of concepts.

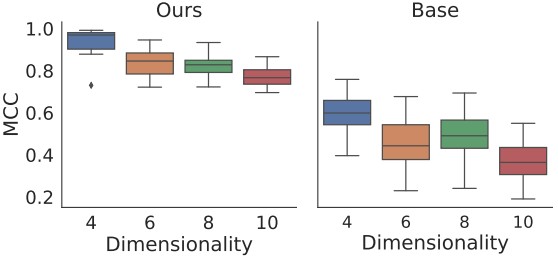

Figure 6: Identification of all concepts w.r.t. different numbers of concepts.

standard Mean Correlation Coefficient (MCC) between the ground-truth and estimated concepts.

The results (Figs. 5 and 6) demonstrate that our models achieve higher MCCs compared to the base model in both settings. This suggests that it is possible to identify hidden concepts from purely observational data without making assumptions about the concept type, functional relationships, or parametric generative models. Meanwhile, our models also provide lower variances across different runs, which further verifies our theoretical findings. As indicated by these results, hidden concepts can be identified up to an element-wise transformation and a permutation under our conditions, while the base model fails to disentangle and recover most concepts from data, further suggesting the necessity of the proposed conditions.

**Real-world datasets.** To assess the applicability of our proposed structural condition in complex practical contexts, we performed experiments on five different real-world datasets, i.e., the Fashion-MNIST (Xiao et al., 2017), EMNIST (Cohen et al., 2017), AnimalFace (Si & Zhu, 2011), Flower102 (Nilsback & Zisserman, 2008), and FFHQ (Karras et al., 2019) datasets. We highlight the identified concepts with the largest standard deviations (SDs) for Fashion-MNIST (Figs. 7 and 8), EMNIST (Fig. 12 in Appx. C.2), Animal-Face (Fig. 9), and FFHQ (Figs. 13, 14, and 15 in Appx. C.2). Each row in the figures shows reconstructed images with the corresponding concept value varying to illustrate its effect. Additionally, the rightmost column features a heat map depicting the absolute pixel differences to visualize the influence. Clearly, the semantics of the identified concepts

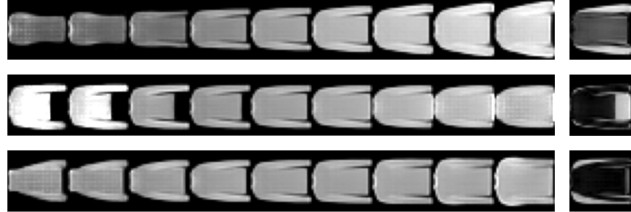

Figure 7: Results on Fashion-MNIST. The rows correspond to different concepts of a pullover: "sleeve length," "torso length," and "shoulder width," respectively.

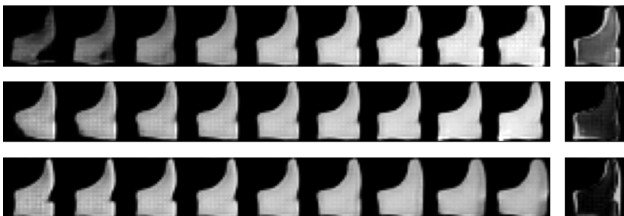

Figure 8: Results on Fashion-MNIST. The rows correspond to different concepts of an ankle boot: "heel height," "ankle width," and "toe box width," respectively.

align with our understanding of the corresponding classes. For Flower102, we test the robustness of the recovered concept by comparing the same concept across different angles and environments. As seen in Fig. 10, the concept can be consistently identified from the same class across various conditions, further supporting our theory. Therefore, these results indicate that hidden concepts can be identified from observational data alone without the need to specify the generative model, underscoring the practical viability. Moreover, certain concepts are inherently entangled across classes (e.g., Figs. 13, 14, and 15 in Appx. C.2), making it impossible to disentangle them individually without additional assumptions. This further highlights the practical significance of our partial identification by local comparison.

## 5. Conclusion

Drawing inspiration from the fundamental cognitive mechanism of learning through comparison, we establish a set of theoretical guarantees for learning concepts in general nonparametric settings. We provide a theoretical framework that potentially explains the impressive empirical successes in many previous works. Specifically, we prove that hidden concepts can be identified up to trivial indeterminacy from diverse classes of observations without any assumptions on the concept types, functional relations, or parametric generating models. Interestingly, even in scenarios where the structural conditions do not universally hold, we can still provide appropriate identifiability for a subset of concepts with sufficient diversity based on the mechanism of local comparison, thereby greatly broadening the applicability of the proposed theory. Furthermore, the connective structure

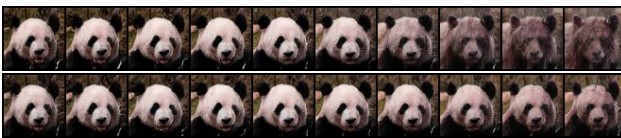

Figure 9: Results on AnimalFace. The rows correspond to "Ursid" and "Monochrome" of a panda, respectively.

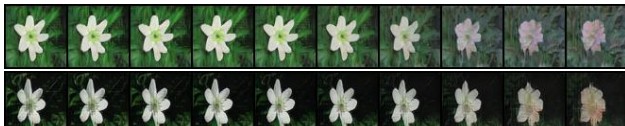

Figure 10: The same concept ("Blooming") consistently identified from different environments in Flower102.

between classes and concepts can also be recovered in a nonparametric manner. As a current limitation, future work involves exploiting the theory to a wider range of practical problems, such as compositional generalization, decision-making, controllable generation, and foundation models.

## Impact Statement

This paper presents work whose goal is to advance the field of Machine Learning. There are many potential societal consequences of our work, none which we feel must be specifically highlighted here.

## Acknowledgment

We appreciate the anonymous reviewers and the area chair for their constructive feedback. We would also like to acknowledge the support from NSF Award No. 2229881, AI Institute for Societal Decision Making (AI-SDM), the National Institutes of Health (NIH) under Contract R01HL159805, and grants from Quris AI, Florin Court Capital, and MBZUAI-WIS Joint Program.

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

# Appendices

## Table of Contents

## A. Summary of Notation

We summarize the key notation used throughout the paper to provide a quick reference for readers.

**Variables and Functions**

- $\mathbf{x} = (\mathbf{x}_1, \ldots, \mathbf{x}_m) \in \mathcal{X} \subseteq \mathbb{R}^m$ : Observed variables.

- $\mathbf{z} = (\mathbf{z}_A, \mathbf{z}_B) \in \mathcal{Z} \subseteq \mathbb{R}^n$, where $n = n_A + n_B$ : Latent concept variables.

- $\mathbf{z}_A \in \mathbb{R}^{n_A}$ : Class-dependent concepts influenced by the classes $\mathbf{c}$.

- $\mathbf{z}_B \in \mathbb{R}^{n_B}$ : Class-independent concepts, unaffected by $\mathbf{c}$.

- $\mathbf{c} = (\mathbf{c}_1, \ldots, \mathbf{c}_u)$ : Class variables represented as vectors, with $u$ classes.

- $f : \mathcal{Z} \to \mathcal{X}$ : Injective generative function mapping latent concepts to observations.

- $\mathbf{z}_A = g(\mathbf{c}, \theta)$ : Class-dependent concept function parameterized by $\mathbf{c}$ and $\theta$ (other factors).

- $\theta$ : Additional influencing factors in the function $g$.

**Probabilities and Densities**

- $p(\mathbf{z} \mid \mathbf{c}) = p(\mathbf{z}_A \mid \mathbf{c})p(\mathbf{z}_B)$ : Conditional density of latent concepts $\mathbf{z}$ given classes $\mathbf{c}$, assuming conditional independence.

- $p(\mathbf{z}_A \mid \mathbf{c}) = \prod_{i=1}^{n_A} p(\mathbf{z}_i \mid M_{i,\cdot} \odot \mathbf{c})$ : Factorized density of class-dependent concepts $\mathbf{z}_A$.

- $\mathbb{E}[\cdot]$ : Expectation operator.

- $\mathbb{P}$ : Probability measure.

**Indices and Sets**

- $A_i$ : Index set of concepts corresponding to class $\mathbf{c}_i$.

- $\mathbf{z}_{A_i}$ : Concepts associated with class $\mathbf{c}_i$.

- $\mathbf{z}_{A_i \setminus A_j}$ : Difference in concept sets between classes $\mathbf{c}_i$ and $\mathbf{c}_j$.

- $\mathcal{I} \subset \{1, \ldots, m\} \times \{1, \ldots, n\}$ : Set of indices for matrix elements.

- $\mathcal{I}_{i,\cdot} = \{j \mid (i,j) \in \mathcal{I}\}$ : Indices corresponding to row $i$ in $\mathcal{I}$.

- $\mathcal{I}_{\cdot,j} = \{i \mid (i,j) \in \mathcal{I}\}$ : Indices corresponding to column $j$ in $\mathcal{I}$.

- $\mathcal{S} \subset \{1, \ldots, n\}$ : Subset of indices.

- $\mathbb{R}^n_{\mathcal{S}} = \{s \in \mathbb{R}^n \mid s_i = 0 \text{ if } i \notin \mathcal{S}\}$ : Subspace of $\mathbb{R}^n$ where components not in $\mathcal{S}$ are zero.

**Matrices and Operations**

- $S \in \mathbb{R}^{a \times b}$ : An arbitrary matrix with the shape $(a, b)$.

- $S_{i,\cdot}, S_{\cdot,j}$ : $i$-th row, $j$-th column of matrix $S$.

- $\mathrm{supp}(S) = \{(i,j) \mid S_{i,j} \neq 0\}$ : Support of matrix $S$.

- $\mathrm{supp}(\mathbf{S}(\Theta)) = \{(i,j) \mid \exists \theta \in \Theta, \mathbf{S}(\theta)_{i,j} \neq 0\}$ : Support of a matrix-valued function $\mathbf{S}(\Theta)$.

- $D_{\mathbf{c}}g$ : Partial derivative of $g$ with respect to class labels $\mathbf{c}$.

- $\mathcal{D} = \mathrm{supp}(D_{\mathbf{c}}g)$ : Support of the Jacobian of $g$ with respect to $\mathbf{c}$.

- $\mathbf{T}$ : Matrix-valued function representing a transformation between $D_{\mathbf{c}}g$ and $D_{\hat{\mathbf{c}}}\hat{g}$.

- $\mathrm{T}$ : A matrix sharing the same support as $\mathbf{T}$.

- $M \in \{0, 1\}^{n_A \times u}$ : Binary structure matrix showing connections between classes and concepts.

- $\odot$ : Element-wise (Hadamard) product.

- $\mathrm{span}\{\cdot\}$ : Linear span of a set of vectors.

- $\mathrm{rank}(\cdot)$ : Rank of a matrix.

**Data and Parameters**

- $\{(\mathbf{x}, \mathbf{c})^{(i)}\}_{i=1}^N$ : Dataset of $N$ samples with observed variables and corresponding classes.

- $\mathcal{M}$ : Mask applied to classes in the dataset.

- $\lambda$ : Regularization parameter used in the estimation objective.

- $\mathbf{R}$ : Regularization term (e.g., $\ell_1$ norm applied to estimated supports).

- $\pi$ : Permutation function used to align estimated concepts.

- $\Theta$ : Parameter space.

### Conventions

- Bold lowercase letters (e.g., $\mathbf{x}$) denote variables, with their realizations denoted by the plain version (e.g., $x$ for $\mathbf{x}$); uppercase letters (e.g., $S$, $M$) denote matrices.

- Calligraphic letters (e.g., $\mathcal{X}$, $\mathcal{Z}$) denote sets or spaces.

- Subscripts with dots denote slicing: $S_{i,\cdot}$ represents the $i$-th row; $S_{\cdot,j}$ represents the $j$-th column; $S_{i,\cdot k}$ denotes the first $k$ dimensions of the $i$-th row and $S_{i,k+1\cdot}$ dentoes the remaining; $S_{\cdot k,j}$ denotes the first $k$ dimensions of the $j$-th column and $S_{k+1\cdot,j}$ denotes the remaining.

- Estimated quantities are denoted with hats (e.g., $\hat{\mathbf{z}}$ for estimated latent concepts).

## B. Proofs

### B.1. Proof of Theorem 1

**Theorem 1** (Learning by pairwise comparison). *Consider two **observationally equivalent** (Defn. 1) models $(g, p_{\mathbf{z}}, M)$ and $(\hat{g}, p_{\hat{\mathbf{z}}}, \hat{M})$ as in Sec. 2. Suppose, with an $\ell_0$ regularization on $D_{\mathbf{c}}\hat{g}$ ($|\hat{\mathcal{D}}| \leq |\mathcal{D}|$), there exist a set of points $\{(c,\theta)^{(\ell)}\}_{\ell=1}^{|\mathcal{D}_{\cdot,i}|}$, such that*

1. *$\{D_{\mathbf{c}}g((c,\theta)^{(\ell)})_{\cdot,i}\}_{\ell=1}^{|\mathcal{D}_{\cdot,i}|}$, are linearly independent;*
2. *$\left[\mathbf{T}D_{\mathbf{c}}g((c,\theta)^{(\ell)})\right]_{\cdot,i} \in \mathbb{R}^{n_A}_{\hat{\mathcal{D}}_{\cdot,i}}$.*

*Then for any pair of classes $\mathbf{c}_i$ and $\mathbf{c}_j$, there exists a permutation $\pi$ such that the unique concepts of each class are disentangled with other concepts, i.e., $\frac{\partial \hat{\mathbf{z}}_{\pi(A_i \setminus A_j)}}{\partial \mathbf{z}_{A_j}}$ and $\frac{\partial \hat{\mathbf{z}}_{\pi(A_j \setminus A_i)}}{\partial \mathbf{z}_{A_i}}$ equal to zero.*

*Proof.* Because of the observational equivalence, we have

$$p_{\hat{\mathbf{x}}|\mathbf{c}} = p_{\mathbf{x}|\mathbf{c}} \Rightarrow p_{\hat{f}(\hat{\mathbf{z}})|\mathbf{c}} = p_{f(\mathbf{z})|\mathbf{c}}. \tag{4}$$

Based on the change-of-variable formula, and the invertibility of the generating function $f$, there exists an invertible function $t := \hat{f}^{-1} \circ f$ such that $\hat{\mathbf{z}} = t(\mathbf{z})$. By taking the derivatives of both sides w.r.t. $\mathbf{c}$, we have

$$D_{\mathbf{c}}\hat{g} = \mathbf{T}D_{\mathbf{c}}g, \tag{5}$$

where $\mathbf{T}$ is matrix-value function that is invertible. According to the assumption, the span is nondegenerate in the sense that

$$\begin{bmatrix} D_{\mathbf{c}}g((c,\theta)^{(1)})_{\cdot,j} \\ D_{\mathbf{c}}g((c,\theta)^{(2)})_{\cdot,j} \\ \vdots \\ D_{\mathbf{c}}g((c,\theta)^{(|\mathcal{D}_{\cdot,j}|)})_{\cdot,j} \end{bmatrix} \tag{6}$$

are linearly independent. Then we can construct an one-hot vector $e_{i_0} \in \mathbb{R}^{n_A}_{\mathcal{D}_{\cdot,j}}$ for any $i_0 \in \mathcal{D}_{\cdot,j}$ as a linear combination of vectors $\{D_{\mathbf{c}}g((c,\theta)^{(\ell)})_{\cdot,j}\}_{\ell=1}^{|\mathcal{D}_{\cdot,j}|}$, i.e., $e_{i_0} = \sum_{\ell \in \mathcal{D}_{\cdot,j}} \beta_\ell D_{\mathbf{c}}g((c,\theta)^{(\ell)})_{\cdot,j}$, where $\beta_\ell$ denotes some coefficient. Note that we define $\mathcal{D}$ as the support of $D_{\mathbf{c}}g$. We also define $\mathrm{T}$ as a matrix with the same support of $\mathbf{T}$ in $D_{\mathbf{c}}\hat{g} = \mathbf{T}D_{\mathbf{c}}g$. Then we have

$$\mathrm{T}_{\cdot,i_0} = \mathrm{T}e_{i_0} = \sum_{\ell \in \mathcal{D}_{\cdot,j}} \beta_\ell \mathrm{T}D_{\mathbf{c}}g((c,\theta)^{(\ell)})_{\cdot,j}. \tag{7}$$

According to the assumption, we have

$$\mathrm{T}D_{\mathbf{c}}g((c,\theta)^{(\ell)})_{\cdot,j} \in \mathbb{R}^{n_A}_{\hat{\mathcal{D}}_{\cdot,j}}. \tag{8}$$

Therefore, Eq. (7) implies $\mathrm{T}_{\cdot,i_0} \in \mathbb{R}^{n_A}_{\hat{\mathcal{D}}_{\cdot,j}}$, which is equivalent to

$$\forall i \in \mathcal{D}_{\cdot,j}, \mathrm{T}_{\cdot,i_0} \in \mathbb{R}^{n_A}_{\hat{\mathcal{D}}_{\cdot,j}}. \tag{9}$$

This further indicates

$$\forall (i,j) \in \mathcal{D}, \mathcal{T}_{.,i} \times \{j\} \subset \hat{\mathcal{D}}. \tag{10}$$

Since $\mathbf{T}$ is invertible, we have

$$\det(\mathbf{T}) = \sum_{\sigma \in \mathcal{S}_{n_A}} \left( \operatorname{sgn}(\sigma) \prod_{j=1}^{n_A} \mathbf{T}_{\sigma(j),j} \right) \neq 0, \tag{11}$$

where $\mathcal{S}_{n_A}$ is a set of $n_A$-permutations. Then there must exist at least one non-zero term in the summation, which indicates that

$$\exists \sigma \in \mathcal{S}_{n_A}, \ \forall j \in \{1, \ldots, n_A\}, \ \operatorname{sgn}(\sigma) \prod_{j=1}^{n_A} \mathbf{T}_{\sigma(j),j} \neq 0. \tag{12}$$

Clearly, there cannot be any term in the product that equals zero, so we have

$$\exists \sigma \in \mathcal{S}_{n_A}, \ \forall j \in \{1, \ldots, n_A\}, \mathbf{T}_{\sigma(j),j} \neq 0. \tag{13}$$

Denote $\mathcal{T}$ as the support of $\mathbf{T}$. Thus, it follows that

$$\forall i \in \{1, \ldots, n_A\}, \sigma(i) \in \mathcal{T}_{.,i}. \tag{14}$$

Then it yields

$$\forall (i,j) \in \mathcal{D}, (\sigma(i), j) \in \mathcal{T}_{.,i} \times \{j\}. \tag{15}$$

Because of Eq. (10), we have

$$\forall (i,j) \in \mathcal{D}, (\sigma(i), j) \in \hat{\mathcal{D}}. \tag{16}$$

Let us denote $\tilde{\pi}(\mathcal{D})$ as a row permutation of $\mathcal{D}$, where $\forall (i,j) \in \mathcal{D}$, there must be

$$(\sigma(i), j) \in \tilde{\pi}(\mathcal{D}) \tag{17}$$

and

$$|\tilde{\pi}(\mathcal{D})| = |\mathcal{D}|. \tag{18}$$

Furthermore, Eq. (16) indicates that

$$\tilde{\pi}(\mathcal{D}) \subset \hat{\mathcal{D}}, \tag{19}$$

We have the following relation based on the sparsity regularization:

$$|\hat{\mathcal{D}}| \leq |\mathcal{D}|. \tag{20}$$

Therefore, we have the following relation:

$$|\tilde{\pi}(\mathcal{D})| = |\mathcal{D}| \geq |\hat{\mathcal{D}}|. \tag{21}$$

Together with Eq. (19), it follows that

$$\hat{\mathcal{D}} = \tilde{\pi}(\mathcal{D}). \tag{22}$$

Let us denote the permutation indeterminacy in our goal as $\pi$ s.t.

$$\hat{\mathcal{D}} := \{(\pi(i), j) \mid (i,j) \in \mathcal{D}\}. \tag{23}$$

Given two classes $\mathbf{c}_i$ and $\mathbf{c}_j$, for any $\mathbf{z}_k \in \mathbf{z}_{A_i}$, we have

$$(k, i) \in \mathcal{D}. \tag{24}$$

Because of Eq. (10), this further implies

$$\mathcal{T}_{.,k} \times \{i\} \in \hat{\mathcal{D}}. \tag{25}$$

For any $\pi(v)$ where $\mathbf{z}_v \in \mathbf{z}_{A_j \setminus A_i}$, suppose we have

$$(\pi(v), k) \in \mathcal{T}, \tag{26}$$

which is equivalent to

$$\pi(v) \in \mathcal{T}_{\cdot,k}. \tag{27}$$

Then according to Eq. (25), we have

$$(\pi(v), i) \in \mathcal{T}_{\cdot,k} \times \{i\} \in \hat{\mathcal{D}}. \tag{28}$$

Based on Eq. (23), Eq. (28) is equivalent to

$$(v, i) \in \mathcal{D}, \tag{29}$$

which indicates a contradiction since $\mathbf{z}_v \in \mathbf{z}_{A_j \setminus A_i}$.

As a result, there must be $(\pi(v), k) \notin \mathcal{T}$. Similarly, for any $\mathbf{z}_u \in \mathbf{z}_{A_j}$, we can also show by contradiction that there must be $(\pi(u), j) \notin \mathcal{T}$. Therefore, for any two classes $\mathbf{c}_i$ and $\mathbf{c}_j$, there exists a permutation $\pi$ that

$$\frac{\partial \hat{\mathbf{z}}_{\pi(A_i \setminus A_j)}}{\partial \mathbf{z}_{A_j}} = 0. \tag{30}$$

Similarly, there is also

$$\frac{\partial \hat{\mathbf{z}}_{\pi(A_j \setminus A_i)}}{\partial \mathbf{z}_{A_i}} = 0, \tag{31}$$

which is the goal. $\qquad\qquad\square$

## B.2. Proof of Corollary 1

**Corollary 1** (Learning by local comparison). *Consider two **observationally equivalent** (Defn. 1) models $(g, p_{\mathbf{z}}, M)$ and $(\hat{g}, p_{\hat{\mathbf{z}}}, \hat{M})$ as in Sec. 2. Suppose that the assumptions in Thm. 1 hold. Then, for a set of classes $\mathbf{c}_I$ and its corresponding concept sets $\mathbf{z}_{A_I}$ with a set of indices $I$, there exists a permutation $\pi$ such that the unique concepts for the class $\mathbf{c}_i$ are disentangled with concepts associated with other classes, i.e., $\frac{\partial \hat{\mathbf{z}}_{\pi(A_i \setminus A_{I \setminus i})}}{\mathbf{z}_{A_{I \setminus i}}} = 0$.*

*Proof.* Because all assumptions in Theorem 1 hold, according to the proof of it, we know that, for a row permutation of $\mathcal{D}$, i.e., $\tilde{\pi}(\mathcal{D})$ where

$$\tilde{\pi}(\mathcal{D}) := \{(\sigma(i), j) | (i, j) \in \mathcal{D}\}. \tag{32}$$

There must be a relationship that

$$\hat{\mathcal{D}} = \tilde{\pi}(\mathcal{D}). \tag{33}$$

Then we want to show that, there exists a permutation $\pi$ that

$$\frac{\partial \hat{\mathbf{z}}_{\pi(A_i \setminus A_{I \setminus i})}}{\mathbf{z}_{A_{I \setminus i}}} = 0. \tag{34}$$

For any $z_k \in \mathbf{z}_{A_{I \setminus i}}$ and its corresponding class $\mathbf{c}_q \in \mathbf{c}_I$ and $q \neq i$, we have

$$(k, q) \in \mathcal{D}. \tag{35}$$

According to the proof of Theorem 1, we have

$$\mathrm{T}D_{\mathbf{c}}g((c, \theta)^{(\ell)})_{\cdot,j} \in \mathbb{R}^{n_A}_{\hat{\mathcal{D}}_{\cdot,j}}. \tag{36}$$

Therefore, Eq. (35) further indicates that

$$\mathcal{T}_{\cdot,k} \times \{q\} \in \hat{\mathcal{D}}, \tag{37}$$

where $\mathcal{T}$ denotes the support of $\mathrm{T}$. Define the permutation $\pi$ as

$$\hat{\mathcal{D}} := \{(\pi(i), j) \mid (i, j) \in \mathcal{D}\}. \tag{38}$$

Then we consider any $\pi(v)$ where we have

$$\mathbf{z}_v \in \mathbf{z}_{A_i \setminus A_{I \setminus i}}. \tag{39}$$

Suppose we have

$$(\pi(v), k) \in \mathcal{T}. \tag{40}$$

This also implies that

$$\pi(v) \in \mathcal{T}_{\cdot, k}. \tag{41}$$

Based on Eq. (37), we further have

$$(\pi(v), q) \in \mathcal{T}_{\cdot, k} \times \{q\} \in \hat{\mathcal{D}}. \tag{42}$$

According to the definition of $\hat{\mathcal{D}}$, this is equivalent to

$$(v, q) \in \mathcal{D}, \tag{43}$$

Because $\mathbf{z}_v \in \mathbf{z}_{A_i \setminus A_{I \setminus i}}$, the above equation indicates that there must be $\mathbf{c}_q = \mathbf{c}_i$. which is a contradiction since $q \neq i$. Therefore, we have

$$(\pi(v), k) \notin \mathcal{T}. \tag{44}$$

This implies that there exists a permutation $\pi$ that

$$\frac{\partial \hat{\mathbf{z}}_{\pi(A_i \setminus A_{I \setminus i})}}{\mathbf{z}_{A_{I \setminus i}}} = 0. \tag{45}$$

This completes the proof. $\qquad\square$

### B.3. Proof of Theorem 2

**Theorem 2** (Learning by global comparison). *Consider two models $(g, p_{\mathbf{z}}, M)$ and $(\hat{g}, p_{\hat{\mathbf{z}}}, \hat{M})$ as in Sec. 2. Under the assumptions in Thm. 1 and Assump. 1, suppose that for any set $A_{\mathbf{z}} \subseteq \mathcal{Z}$ with non-zero probability measure and which cannot be expressed as $B_{\mathbf{z}_B} \times \mathbf{z}_A$ for any $B_{\mathbf{z}_B} \subseteq \mathcal{Z}_B$, there exist two values of $\mathbf{c}$, i.e., $c^{(k)}$ and $c^{(v)}$ (which may vary across different $A_{\mathbf{z}}$), that*

$$\int_{\mathbf{z} \in A_{\mathbf{z}}} p(\mathbf{z} \mid c^{(k)}) d\mathbf{z} \neq \int_{\mathbf{z} \in A_{\mathbf{z}}} p(\mathbf{z} \mid c^{(v)}) d\mathbf{z}.$$

*If $(g, p_{\mathbf{z}}, M)$ and $(\hat{g}, p_{\hat{\mathbf{z}}}, \hat{M})$ are **observationally equivalent** (Defn. 1), then:*

1. *(Element-wise Identifiability) For any $\mathbf{z}_i \in \mathbf{z}_A$, there exists a permutation $\pi$ and invertible functions $h_i : \mathbb{R} \to \mathbb{R}$ s.t. $\hat{\mathbf{z}}_i = h_i(\mathbf{z}_{\pi(i)})$;*

2. *(Block-wise Identifiability) For $\mathbf{z}_B$, there exists an invertible function $h : \mathbb{R}^{n_B} \to \mathbb{R}^{n_B}$ s.t. $\hat{\mathbf{z}}_B = h(\mathbf{z}_B)$.*

*Proof.* Because of the observational equivalence, we have

$$p_{\hat{\mathbf{x}} \mid \mathbf{c}} = p_{\mathbf{x} \mid \mathbf{c}} \Rightarrow p_{\hat{f}(\hat{\mathbf{z}}) \mid \mathbf{c}} = p_{f(\mathbf{z}) \mid \mathbf{c}}. \tag{46}$$

Based on the change-of-variable formula, and the invertibility of the generating function $f$, there exists an invertible function $h := \hat{f}^{-1} \circ f$ such that $\hat{\mathbf{z}} = h(\mathbf{z})$. Using the chain rule, the derivative of $\hat{g}$ with respect to $\mathbf{c}$ can be expressed as:

$$D_{\mathbf{c}} \hat{g} = D_{\mathbf{z}} h D_{\mathbf{c}} g. \tag{47}$$

The Jacobian of $h$ can be written as:

$$D_{\mathbf{z}} h = \left[ \begin{array}{c|c} \frac{\partial \hat{\mathbf{z}}_A}{\partial \mathbf{z}_A} & \frac{\partial \hat{\mathbf{z}}_A}{\partial \mathbf{z}_B} \\ \hline \frac{\partial \hat{\mathbf{z}}_B}{\partial \mathbf{z}_A} & \frac{\partial \hat{\mathbf{z}}_B}{\partial \mathbf{z}_B} \end{array} \right]. \tag{48}$$

According to steps 1, 2, and 3 in the proof of Theorem 4.2 in Kong et al. (2022), the bottom-left block of $D_{\mathbf{z}} h$, i.e., $D_{\mathbf{z}} h_{n_A + 1 \cdot, \cdot n_A}$, consists of only zero entries. As a result, the Jacobian is equivalent to:

$$D_{\mathbf{z}} h = \left[ \begin{array}{c|c} \frac{\partial \hat{\mathbf{z}}_A}{\partial \mathbf{z}_A} & \frac{\partial \hat{\mathbf{z}}_A}{\partial \mathbf{z}_B} \\ \hline \mathbf{0} & \frac{\partial \hat{\mathbf{z}}_B}{\partial \mathbf{z}_B} \end{array} \right]. \tag{49}$$

Since $h$ is invertible, the determinant of $D_{\mathbf{z}}h$ is non-zero. Together with the structure of the Jacobian matrix, we have

$$\det(D_{\mathbf{z}}h) = \det(\frac{\partial \hat{\mathbf{z}}_A}{\partial \mathbf{z}_A}) \det(\frac{\partial \hat{\mathbf{z}}_B}{\partial \mathbf{z}_B}), \tag{50}$$

which further implies

$$\det(\frac{\partial \hat{\mathbf{z}}_A}{\partial \mathbf{z}_A}) \neq 0, \tag{51}$$

$$\det(\frac{\partial \hat{\mathbf{z}}_B}{\partial \mathbf{z}_B}) \neq 0. \tag{52}$$

Since $\det(\frac{\partial \hat{\mathbf{z}}_B}{\partial \mathbf{z}_B}) \neq 0$ and $\frac{\partial \hat{\mathbf{z}}_B}{\partial \mathbf{z}_A} = 0$, there exists an invertible function $h_B : \mathbf{z}_B \to \hat{\mathbf{z}}_B$ s.t.,

$$\hat{\mathbf{z}}_B = h_B(\mathbf{z}_B). \tag{53}$$

Since $\hat{\mathbf{z}}_A$ is independent of $\hat{\mathbf{z}}_B$ and $\hat{\mathbf{z}}_B = h_B(\mathbf{z}_B)$, we further have

$$\frac{\partial \hat{\mathbf{z}}_A}{\partial \mathbf{z}_B} = 0. \tag{54}$$

Then the Jacobian can be represented as

$$D_{\mathbf{z}}h = \left[ \begin{array}{c|c} \frac{\partial \hat{\mathbf{z}}_A}{\partial \mathbf{z}_A} & \mathbf{0} \\ \hline \mathbf{0} & \frac{\partial \hat{\mathbf{z}}_B}{\partial \mathbf{z}_B} \end{array} \right]. \tag{55}$$

Thus, $\hat{\mathbf{z}}_B$ is identifiable up to a block-wise invertible transformation, and we have

$$\begin{cases} \frac{\partial \hat{\mathbf{z}}_i}{\partial \mathbf{z}_j} = 0 & i \in \{1, \ldots, n_A\}, j \in \{n_A + 1, \ldots, n\}, \\ \frac{\partial \hat{\mathbf{z}}_k}{\partial \mathbf{z}_v} = 0 & k \in \{n_A + 1, \ldots, n\}, v \in \{1, \ldots, n_A\}. \end{cases} \tag{56}$$

This implies that

$$D_{\mathbf{c}}\hat{g} = D_{\mathbf{z}}h_{\cdot n_A, \cdot n_A} D_{\mathbf{c}}g. \tag{57}$$

According to the assumption, we have a set of linearly independent vectors as follows

$$\begin{bmatrix} D_{\mathbf{c}}g((c,\theta)^{(1)})_{\cdot,j} \\ D_{\mathbf{c}}g((c,\theta)^{(2)})_{\cdot,j} \\ \vdots \\ D_{\mathbf{c}}g((c,\theta)^{(|\mathcal{D}_{\cdot,j}|)})_{\cdot,j} \end{bmatrix} \tag{58}$$

Then we can construct an one-hot vector $e_{i_0} \in \mathbb{R}^{n_A}_{\mathcal{D}_{\cdot,j}}$ for any $i_0 \in \mathcal{D}_{\cdot,j}$ as a linear combination of vectors $\{D_{\mathbf{c}}g((c,\theta)^{(\ell)})_{\cdot,j}\}_{\ell=1}^{|\mathcal{D}_{\cdot,j}|}$, i.e., $e_{i_0} = \sum_{\ell \in \mathcal{D}_{\cdot,j}} \beta_\ell D_{\mathbf{c}}g((c,\theta)^{(\ell)})_{\cdot,j}$, where $\beta_\ell$ denotes some coefficient. Note that we define T as a matrix with the same support of $\hat{\mathbf{T}}$ in $D_{\mathbf{c}}\hat{g} = \mathbf{T}D_{\mathbf{c}}g$, where $\mathbf{T}$ is a matrix-valued function. Then we have

$$\mathbf{T}_{\cdot,i_0} = \mathbf{T}e_{i_0} = \sum_{\ell \in \mathcal{D}_{\cdot,j}} \beta_\ell \mathbf{T} D_{\mathbf{c}}g((c,\theta)^{(\ell)})_{\cdot,j}. \tag{59}$$

According to the assumption, we have

$$\mathbf{T} D_{\mathbf{c}}g((c,\theta)^{(\ell)})_{\cdot,j} \in \mathbb{R}^{n_A}_{\hat{\mathcal{D}}_{\cdot,j}}. \tag{60}$$

Therefore, Eq. (59) implies $\mathbf{T}_{\cdot,i_0} \in \mathbb{R}^{n_A}_{\hat{\mathcal{D}}_{\cdot,j}}$, which is equivalent to

$$\forall i \in \mathcal{D}_{\cdot,j}, \mathbf{T}_{\cdot,i_0} \in \mathbb{R}^{n_A}_{\hat{\mathcal{D}}_{\cdot,j}}. \tag{61}$$

This further indicates

$$\forall (i,j) \in \mathcal{D}, \mathcal{T}_{\cdot,i} \times \{j\} \subset \hat{\mathcal{D}}, \tag{62}$$

where $\mathcal{T}$ denotes the support of $\mathbf{T}$. Since $\mathbf{T}$ is invertible, we have

$$\det(\mathbf{T}) = \sum_{\sigma \in \mathcal{S}_{n_A}} \left( \operatorname{sgn}(\sigma) \prod_{j=1}^{n_A} \mathbf{T}_{\sigma(j),j} \right) \neq 0, \tag{63}$$

where $\mathcal{S}_{n_A}$ is a set of $n_A$-permutations. Then there must exist at least one non-zero term in the summation, which indicates that

$$\exists \sigma \in \mathcal{S}_{n_A}, \, \forall j \in \{1, \ldots, n_A\}, \, \operatorname{sgn}(\sigma) \prod_{j=1}^{n_A} \mathbf{T}_{\sigma(j),j} \neq 0. \tag{64}$$

Clearly, there cannot be any term in the product that equals zero, so we have

$$\exists \sigma \in \mathcal{S}_{n_A}, \, \forall j \in \{1, \ldots, n_A\}, \mathbf{T}_{\sigma(j),j} \neq 0. \tag{65}$$

Thus, it follows that

$$\forall i \in \{1, \ldots, n_A\}, \sigma(i) \in \mathcal{T}_{\cdot,i}. \tag{66}$$

Then it yields

$$\forall (i, j) \in \mathcal{D}, (\sigma(i), j) \in \mathcal{T}_{\cdot,i} \times \{j\}. \tag{67}$$

Because of Eq. (62), we have

$$\forall (i, j) \in \mathcal{D}, (\sigma(i), j) \in \hat{\mathcal{D}}. \tag{68}$$

Let us denote $\tilde{\pi}(\mathcal{D})$ as a row permutation of $\mathcal{D}$, where $\forall (i, j) \in \mathcal{D}$, there must be

$$(\sigma(i), j) \in \tilde{\pi}(\mathcal{D}), \tag{69}$$

and

$$|\tilde{\pi}(\mathcal{D})| = |\mathcal{D}|. \tag{70}$$

Eq. 68 indicates that

$$\tilde{\pi}(\mathcal{D}) \subset \hat{\mathcal{D}}. \tag{71}$$

According to the sparsity regularization, we have the following relation based on the sparsity regularization:

$$|\hat{\mathcal{D}}| \leq |\mathcal{D}|. \tag{72}$$

Therefore, we have

$$|\tilde{\pi}(\mathcal{D})| = |\mathcal{D}| \geq |\hat{\mathcal{D}}|. \tag{73}$$

Together with Eq. (71), it follows that

$$\hat{\mathcal{D}} = \tilde{\pi}(\mathcal{D}). \tag{74}$$

Let us denote the permutation indeterminacy in our goal as $\pi$ s.t.

$$\hat{\mathcal{D}} := \{(\pi(i), j) \mid (i, j) \in \mathcal{D}\}. \tag{75}$$

For a latent concept $\mathbf{z}_i$, according to the structural diversity assumption (Assump. 1), there exists a set of column indices $J$, where $M_{i,J}$ only has one non-zero entry. Let us denote that non-zero entry as $M_{i,j}$. Since $M$ is a binary matrix with the support $\mathcal{D}$, we have $(i, j) \in \mathcal{D}$ and $(i, k) \notin \mathcal{D}$ for any $k \in J \setminus j$.

Then, according to the assumption, for any other concept $\mathbf{z}_v$ where $v \neq i$, there must be a class $\mathbf{c}_q$ s.t. $q \in J \setminus j$ s.t.

$$(v, q) \in \mathcal{D}. \tag{76}$$

Because of Eq. (62), it follows that

$$\mathcal{T}_{\cdot,v} \times \{q\} \in \hat{\mathcal{D}}. \tag{77}$$

For any $\pi(i)$, suppose we have

$$(\pi(i), v) \in \mathcal{T}, \tag{78}$$

which is equivalent to

$$\pi(i) \in \mathcal{T}_{\cdot,v}. \tag{79}$$

Then according to Eq. (77), we have

$$(\pi(i), q) \in \mathcal{T}_{\cdot,v} \times \{q\} \in \hat{\mathcal{D}}. \tag{80}$$

Based on Eq. (75), Eq. (80) is equivalent to

$$(i, q) \in \mathcal{D}. \tag{81}$$

This is a contradiction since $(i, q) \notin \mathcal{D}$ for any $q \in J \setminus j$. Thus, for any $i \in \{1, \ldots, n_A\}$ and $v \in \{1, \ldots, n_A\} \setminus \{i\}$, there must be

$$(\pi(i), v) \notin \mathcal{T}. \tag{82}$$

Because $\mathcal{T}$ is invertible, all row must have at least one non-zero entry. Thus, Eq. (82) further implies

$$(\pi(i), i) \in \mathcal{T}. \tag{83}$$

Combining both Eqs. (82) and (83) for each $i \in \{1, \ldots, n_A\}$, the transformation between $\hat{\mathbf{z}}_A$ and $\mathbf{z}_A$ must be a composition of an element-wise invertible transformation and a permutation, which is our goal. □

## B.4. Proof of Proposition 1

**Proposition 1** (Learning class-independent concepts; **Informal**). *Consider two **observationally equivalent** (Defn. 1) models $(g, p_{\mathbf{z}}, M)$ and $(\hat{g}, p_{\hat{\mathbf{z}}}, \hat{M})$ as in Sec. 2. Under the assumptions in Thm. 2, further assume structural conditions on the connective structure between $\mathbf{z}_B$ and $\mathbf{x}$ and an $\ell_0$ regularization. Then for any $\mathbf{z}_i \in \mathbf{z}$, there exists a permutation $\pi$ and invertible functions $h_i : \mathbb{R} \to \mathbb{R}$ s.t. $\hat{\mathbf{z}}_i = h_i(\mathbf{z}_{\pi(i)})$.*

We first present its formal version. For brevity, let $\mathcal{F}$ and $\hat{\mathcal{F}}$ denote the support of the Jacobian $D_{\mathbf{z}}f$ and $D_{\hat{\mathbf{z}}}\hat{f}$, respectively. Additionally, $\mathrm{T}_f$ refers to a matrix with the same support of $\mathbf{T}_f$ in $D_{\hat{\mathbf{z}}}\hat{f} = D_{\mathbf{z}}f\mathbf{T}_f$, where $\mathbf{T}_f$ is a matrix-valued function. Generally, the condition on the structure $\mathrm{supp}(D_{\mathbf{z}}f)$ encourages sparsity in the Jacobian of the generating function $f$. As verified empirically in previous work (Zheng & Zhang, 2023), this condition is likely to hold in our setting where the number of observed variables $\mathbf{x}$ exceeds the number of class-independent concepts $\mathbf{z}_B$.

**Proposition 1** (Learning class-independent concepts; **Formal**). *Consider two **observationally equivalent** (Defn. 1) models $(g, p_{\mathbf{z}}, M)$ and $(\hat{g}, p_{\hat{\mathbf{z}}}, \hat{M})$ generated as in Sec. 2. Under the assumptions in Thm. 2, further suppose that, for all $\mathbf{z}_i \in \mathbf{z}_B$, with an $\ell_0$ regularization on $D_{\hat{\mathbf{z}}}\hat{f}$ ($|\hat{\mathcal{F}}| \leq |\mathcal{F}|$), there exists $\mathcal{C}_i$ s.t. $\bigcap_{k \in \mathcal{C}_i} \mathrm{supp}(D_{\mathbf{z}_i}f)_{i,n_A+1\cdot} = \{i\}$. Meanwhile, for each $i \in \{n_A + 1, \ldots, n\}$, there exist $\{\mathbf{z}^{(\ell)}\}_{\ell=1}^{|\mathcal{F}_{i,n_A+1\cdot}|}$ s.t. $\{D_{\mathbf{z}}f(\mathbf{z}^{(\ell)})_{i,n_A+1\cdot}\}_{\ell=1}^{|\mathcal{F}_{i,n_A+1\cdot}|}$ are linearly independent, and $\left[D_{\mathbf{z}}f(\mathbf{z}^{(\ell)})\mathrm{T}_f\right]_{i,n_A+1\cdot} \in \mathbb{R}^{n_B}_{\hat{\mathcal{F}}_{i,n_A+1\cdot}}$. Then for any $\mathbf{z}_i \in \mathbf{z}$, there exists a permutation $\pi$ and invertible functions $h_i : \mathbb{R} \to \mathbb{R}$ such that $\hat{\mathbf{z}}_i = h_i(\mathbf{z}_{\pi(i)})$ for all $i$.*

*Proof.* Because of the observational equivalence, we have

$$p_{\hat{\mathbf{x}}|\mathbf{c}} = p_{\mathbf{x}|\mathbf{c}} \Rightarrow p_{\hat{f}(\hat{\mathbf{z}})|\mathbf{c}} = p_{f(\mathbf{z})|\mathbf{c}}. \tag{84}$$

Based on the change-of-variable formula, and the invertibility of the generating function $f$, there exists an invertible function $h := \hat{f}^{-1} \circ f$ such that $\hat{\mathbf{z}} = h(\mathbf{z})$. According to the proof in Theorem 2, the Jacobian of $h$ w.r.t. $\mathbf{z}$ is as follows:

$$D_{\mathbf{z}}h = \begin{bmatrix} \dfrac{\partial \hat{\mathbf{z}}_A}{\partial \mathbf{z}_A} & \mathbf{0} \\ \hline \mathbf{0} & \dfrac{\partial \hat{\mathbf{z}}_B}{\partial \mathbf{z}_B} \end{bmatrix}. \tag{85}$$

At the same time, by using the chain rule on $h = \hat{f}^{-1} \circ f$, we have

$$D_{\hat{\mathbf{z}}}\hat{f} = D_{\mathbf{z}}f D_{\hat{\mathbf{z}}}h^{-1}, \tag{86}$$

which is equivalent to

$$D_{\hat{\mathbf{z}}}\hat{f}_{\cdot,n_A+1\cdot} = D_{\mathbf{z}}f D_{\hat{\mathbf{z}}}h^{-1}_{\cdot,n_A+1\cdot}. \tag{87}$$

Based on Eq. 85, this further indicates that

$$D_{\hat{\mathbf{z}}}\hat{f}_{\cdot,n_A+1\cdot} = D_{\mathbf{z}}f_{\cdot,n_A+1\cdot}D_{\hat{\mathbf{z}}}h^{-1}{}_{n_A+1\cdot,n_A+1\cdot}. \tag{88}$$

Then, according to the assumption, there must be a set of vectors

$$\begin{bmatrix} D_{\mathbf{z}}f(\mathbf{z}^{(1)})_{i,n_A+1\cdot} \\ D_{\mathbf{z}}f(\mathbf{z}^{(2)})_{i,n_A+1\cdot} \\ \vdots \\ D_{\mathbf{z}}f(\mathbf{z}^{(|\mathcal{F}_{i,n_A+1\cdot}|)})_{i,n_A+1\cdot} \end{bmatrix} \tag{89}$$

that are linearly independent. Then we can construct an one-hot vector $e_{j_0} \in \mathbb{R}^{n_B}_{\mathcal{F}_{i,n_A+1\cdot}}$ for any $j_0 \in \mathcal{F}_{i,n_A+1\cdot}$ as a linear combination of vectors $\{D_{\mathbf{z}}f(\mathbf{z}^{(\ell)})_{i,n_A+1\cdot}\}_{\ell=1}^{|\mathcal{F}_{i,n_A+1\cdot}|}$, i.e.,

$$e_{j_0} = \sum_{\ell \in \mathcal{F}_{i,n_A+1\cdot}} \beta_\ell D_{\mathbf{z}}f(\mathbf{z}^{(\ell)})_{i,n_A+1\cdot}, \tag{90}$$

where $\beta_\ell$ denotes some coefficient. Then we have

$$\mathrm{T}_{fj_0,n_A+1\cdot} = e_{j_0}\mathrm{T}_{f\cdot,n_A+1\cdot} = \sum_{\ell \in \mathcal{D}_{\cdot,j}} \beta_\ell D_{\mathbf{z}}f(\mathbf{z}^{(\ell)})_{i,n_A+1\cdot}\mathrm{T}_{f\cdot,n_A+1\cdot} \in \mathbb{R}^{n_B}_{\hat{\mathcal{F}}_{i,n_A+1\cdot}}. \tag{91}$$

This further implies that, for any $j \in \mathcal{F}_{i,n_A+1\cdot}$, we always have $\mathrm{T}_{fj,\cdot} \in \mathbb{R}^{n_B}_{\hat{\mathcal{F}}_{i,n_A+1\cdot}}$. Thus, we have the connection between support as follows:

$$(i,j) \in \mathcal{F}_{\cdot,n_A+1\cdot}, \{i\} \times \mathcal{T}_{fj,\cdot} \subset \hat{\mathcal{F}}_{\cdot,n_A+1\cdot}. \tag{92}$$

Then, because of the invertibility of $\mathbf{T}_f$, its determinant must not equal to zero, i.e.,

$$\sum_{\sigma \in \mathcal{S}_n} \left( \mathrm{sgn}(\sigma)\prod_{i=1}^{n_B}\mathbf{T}_f(\mathbf{z}^{(\ell)})_{i,\sigma(i)} \right) \neq 0, \tag{93}$$

where $\mathcal{S}$ is the set of $n$-permutations. Therefore, there must be at least one term in the summation that does not equal to zero, i.e.,

$$\exists \sigma \in \mathcal{S}_n, \forall i \in \{1,\dots,n_B\}, \mathrm{sgn}(\sigma)\prod_{i=1}^{n_B}\mathbf{T}_f(\mathbf{z}^{(\ell)})_{i,\sigma(i)} \neq 0. \tag{94}$$

Because $\mathrm{sgn}(\sigma) \neq 0$, every term in the production must not equal to zero, i.e.,

$$\exists \sigma \in \mathcal{S}_n, \forall i \in \{1,\dots,n_B\}, \mathbf{T}_f(\mathbf{z}^{(\ell)})_{i,\sigma(i)} \neq 0. \tag{95}$$

This follows that

$$\forall j \in \{1,\dots,n_B\}, \sigma(j) \in \mathcal{T}_{fj,n_A+1\cdot}. \tag{96}$$

Based on Eq. (92), Eq. (96) further implies that, for any $(i,j) \in \mathcal{F}_{\cdot,n_A+1\cdot}$, we have $(i,\sigma(j)) \in \hat{\mathcal{F}}_{\cdot,n_A+1\cdot}$. Let us denote $\sigma(\mathcal{F}) = \{(i,\sigma(j)) \mid (i,j) \in \mathcal{F}\}$, the above connection implies $\sigma(\mathcal{F}) \subset \hat{\mathcal{F}}$. Together with the sparsity regularization on the estimated Jacobian, we have

$$|\hat{\mathcal{F}}| \leq |\mathcal{F}| \tag{97}$$

Because of the definition of $\sigma(\mathcal{F})$, there must be

$$|\mathcal{F}| = |\sigma(\mathcal{F})|, \tag{98}$$

which follows that

$$|\sigma(\mathcal{F})| \geq |\hat{\mathcal{F}}|. \tag{99}$$

Together with the relation that $\sigma(\mathcal{F}) \subset \hat{\mathcal{F}}$, there must be

$$\hat{\mathcal{F}} = \sigma(\mathcal{F}). \tag{100}$$

Suppose $\mathbf{T}_{\cdot,n_A+1\cdot}$ is not a composition of a permutation matrix and a diagonal matrix, then

$$\exists j_1 \neq j_2, \; \mathcal{T}_{j_1,n_A+1\cdot} \cap \mathcal{T}_{j_2,n_A+1\cdot} \neq \emptyset. \tag{101}$$

Additionally, consider $j_3 \in \{1, \ldots, n_B\}$ for which

$$\sigma(j_3) \in \mathcal{T}_{j_1,n_A+1\cdot} \cap \mathcal{T}_{j_2,n_A+1\cdot}. \tag{102}$$

Since $j_1 \neq j_2$, we can assume $j_3 \neq j_1$ without loss of generality. Based on assumption, there exists $\mathcal{C}_{j_1} \ni j_1$ such that $\bigcap_{i \in \mathcal{C}_{j_1}} \mathcal{F}_{i,n_A+1\cdot} = \{j_1\}$. Because

$$j_3 \notin \{j_1\} = \bigcap_{i \in \mathcal{C}_{j_1}} \mathcal{F}_{i,n_A+1\cdot}, \tag{103}$$

there must exists $i_3 \in \mathcal{C}_{j_1}$ such that

$$j_3 \notin \mathcal{F}_{i_3,n_A+1\cdot}. \tag{104}$$

Since $j_1 \in \mathcal{F}_{i_3,n_A+1\cdot}$, it follows that $(i_3, j_1) \in \mathcal{F}_{\cdot,n_A+1\cdot}$. Therefore, according to Eq. (92), we have

$$\{i_3\} \times \mathcal{T}_{j_1,n_A+1\cdot} \subset \hat{\mathcal{F}}_{\cdot,n_A+1\cdot}. \tag{105}$$

Notice that $\sigma(j_3) \in \mathcal{T}_{j_1,n_A+1\cdot} \cap \mathcal{T}_{j_2,n_A+1\cdot}$ implies

$$(i_3, \sigma(j_3)) \in \{i_3\} \times \mathcal{T}_{j_1,n_A+1\cdot}. \tag{106}$$

Then by Eqs. (105) and (106), we have

$$(i_3, \sigma(j_3)) \in \hat{\mathcal{F}}_{\cdot,n_A+1\cdot}. \tag{107}$$

This further implies $(i_3, j_3) \in \mathcal{F}_{\cdot,n_A+1\cdot}$ by Eq. (100), which contradicts Eq. (104). Therefore, we have proven by contradiction that $\mathbf{T}_{\cdot,n_A+1\cdot}$ is a composition of a permutation matrix and a diagonal matrix, which means that the invariant part $\mathbf{z}_B$ is identifiable up to an element-wise invertible transformation and a permutation. Together with the element-wise identifiability for concepts in the changing part $\mathbf{z}_A$ given by Theorem 2, we have proved that all latent concepts $\mathbf{z} = (\mathbf{z}_A, \mathbf{z}_B)$ are identifiable up to an element-wise invertible transformation and a permutation. $\square$

## B.5. Proof of Proposition 2

**Proposition 2** (Learning class-concept structure). *Consider two **observationally equivalent** (Defn. 1) models $(g, p_{\mathbf{z}}, M)$ and $(\hat{g}, p_{\hat{\mathbf{z}}}, \hat{M})$ as in Sec. 2. Suppose all assumptions in Thm. 1 hold, except Assump. 1. Then $\hat{M} = PM$ for a permutation matrix $P$.*

*Proof.* Because of the observational equivalence, we have

$$p_{\hat{\mathbf{x}}|\mathbf{c}} = p_{\mathbf{x}|\mathbf{c}} \Rightarrow p_{\hat{f}(\hat{\mathbf{z}})|\mathbf{c}} = p_{f(\mathbf{z})|\mathbf{c}}. \tag{108}$$

Based on the change-of-variable formula, and the invertibility of the generating function $f$, there exists an invertible function $h := \hat{f}^{-1} \circ f$ such that $\hat{\mathbf{z}} = h(\mathbf{z})$.

Using the chain rule, the derivative of both sides with respect to $\mathbf{c}$ can be expressed as:

$$D_{\mathbf{c}}\hat{g} = D_{\mathbf{z}}h D_{\mathbf{c}}g. \tag{109}$$

The Jacobian of $h$ can be written as:

$$D_{\mathbf{z}}h = \left[\begin{array}{c|c} \frac{\partial \hat{\mathbf{z}}_A}{\partial \mathbf{z}_A} & \frac{\partial \hat{\mathbf{z}}_A}{\partial \mathbf{z}_B} \\ \hline \frac{\partial \hat{\mathbf{z}}_B}{\partial \mathbf{z}_A} & \frac{\partial \hat{\mathbf{z}}_B}{\partial \mathbf{z}_B} \end{array}\right]. \tag{110}$$

According to steps 1, 2, and 3 in the proof of Theorem 4.2 in Kong et al. (2022), the bottom-left block of $D_{\mathbf{z}}h$, i.e., $D_{\mathbf{z}}h_{n_A+1\cdot,\cdot n_A}$, consists of only zero entries. As a result, the Jacobian is equivalent to:

$$D_{\mathbf{z}}h = \left[\begin{array}{c|c} \frac{\partial \hat{\mathbf{z}}_A}{\partial \mathbf{z}_A} & \frac{\partial \hat{\mathbf{z}}_A}{\partial \mathbf{z}_B} \\ \hline \mathbf{0} & \frac{\partial \hat{\mathbf{z}}_B}{\partial \mathbf{z}_B} \end{array}\right]. \tag{111}$$

Since $h$ is invertible, the determinant of $D_{\mathbf{z}}h$ is non-zero. Together with the structure of the Jacobian matrix, we have

$$\det(D_{\mathbf{z}}h) = \det(\frac{\partial \hat{\mathbf{z}}_A}{\partial \mathbf{z}_A}) \det(\frac{\partial \hat{\mathbf{z}}_B}{\partial \mathbf{z}_B}), \tag{112}$$

which further implies

$$\det(\frac{\partial \hat{\mathbf{z}}_A}{\partial \mathbf{z}_A}) \neq 0, \tag{113}$$

$$\det(\frac{\partial \hat{\mathbf{z}}_B}{\partial \mathbf{z}_B}) \neq 0. \tag{114}$$

Since $\det(\frac{\partial \hat{\mathbf{z}}_B}{\partial \mathbf{z}_B}) \neq 0$ and $\frac{\partial \hat{\mathbf{z}}_B}{\partial \mathbf{z}_A} = 0$, there exists an invertible function $h_B : \mathbf{z}_B \to \hat{\mathbf{z}}_B$ s.t.,

$$\hat{\mathbf{z}}_B = h_B(\mathbf{z}_B). \tag{115}$$

Since $\hat{\mathbf{z}}_A$ is independent of $\hat{\mathbf{z}}_B$ and $\hat{\mathbf{z}}_B = h_B(\mathbf{z}_B)$, we further have

$$\frac{\partial \hat{\mathbf{z}}_A}{\partial \mathbf{z}_B} = 0. \tag{116}$$

Therefore, the Jacobian of $h$ is

$$D_{\mathbf{z}}h = \left[ \begin{array}{c|c} \frac{\partial \hat{\mathbf{z}}_A}{\partial \mathbf{z}_A} & \mathbf{0} \\ \hline \mathbf{0} & \frac{\partial \hat{\mathbf{z}}_B}{\partial \mathbf{z}_B} \end{array} \right]. \tag{117}$$

Note that we have

$$D_{\mathbf{c}}\hat{g} = D_{\mathbf{z}}h D_{\mathbf{c}}g, \tag{118}$$

which is equivalent to

$$D_{\mathbf{c}}\hat{g} = (D_{\mathbf{z}}h D_{\mathbf{c}}g)_{\cdot n_A, \cdot} = D_{\mathbf{z}}h_{\cdot n_A, \cdot} D_{\mathbf{c}}g. \tag{119}$$

Because $\frac{\partial \hat{\mathbf{z}}_i}{\partial \mathbf{z}_k} = 0$ for $i \in \{1, \dots, n_A\}$ and $k \in \{n_A + 1, \dots, n\}$, the upper-right block of $D_{\mathbf{z}}h$, i.e., $D_{\mathbf{z}}h_{\cdot n_A, n_A+1 \cdot}$, consists of only zero entries. It further indicates that

$$D_{\mathbf{c}}\hat{g} = D_{\mathbf{z}}h_{\cdot n_A, \cdot n_A} D_{\mathbf{c}}g. \tag{120}$$

According to the assumption, we have

$$\begin{bmatrix} D_{\mathbf{c}}g((c,\theta)^{(1)})_{\cdot,j} \\ D_{\mathbf{c}}g((c,\theta)^{(2)})_{\cdot,j} \\ \vdots \\ D_{\mathbf{c}}g((c,\theta)^{(|\mathcal{D}_{\cdot,j}|)})_{\cdot,j} \end{bmatrix} \tag{121}$$

as a set of linearly independent vectors.

Then we can construct an one-hot vector $e_{i_0} \in \mathbb{R}^{n_A}_{\mathcal{D}_{\cdot,j}}$ for any $i_0 \in \mathcal{D}_{\cdot,j}$ as a linear combination of vectors $\{D_{\mathbf{c}}g((c,\theta)^{(\ell)})_{\cdot,j}\}_{\ell=1}^{|\mathcal{D}_{\cdot,j}|}$, i.e., $e_{i_0} = \sum_{\ell \in \mathcal{D}_{\cdot,j}} \beta_\ell D_{\mathbf{c}}g((c,\theta)^{(\ell)})_{\cdot,j}$, where $\beta_\ell$ denotes some coefficient. Then we have

$$\mathrm{T}_{\cdot,i_0} = \mathrm{T}e_{i_0} = \sum_{\ell \in \mathcal{D}_{\cdot,j}} \beta_\ell \mathrm{T} D_{\mathbf{c}}g((c,\theta)^{(\ell)})_{\cdot,j}. \tag{122}$$

Note that we define $\mathcal{D}$ as the support of $D_{\mathbf{c}}g$. Additionally, we define $\mathrm{T}$ as a matrix that share the same support as $\mathbf{T}$ in the equation $D_{\mathbf{c}}\hat{g} = \mathbf{T} D_{\mathbf{c}}g$, where $\mathbf{T}$ is a matrix-valued function and $\mathrm{T} \in \mathcal{T}$.

According to the assumption, we have

$$\mathrm{T} D_{\mathbf{c}}g((c,\theta)^{(\ell)})_{\cdot,j} \in \mathbb{R}^{n_A}_{\hat{\mathcal{D}}_{\cdot,j}}. \tag{123}$$

Therefore, Eq. (122) implies $\mathrm{T}_{\cdot,i_0} \in \mathbb{R}^{n_A}_{\hat{\mathcal{D}}_{\cdot,j}}$, which is equivalent to

$$\forall i_0 \in \mathcal{D}_{\cdot,j}, \mathrm{T}_{\cdot,i_0} \in \mathbb{R}^{n_A}_{\hat{\mathcal{D}}_{\cdot,j}}. \tag{124}$$

This further indicates
$$\forall (i,j) \in \mathcal{D}, \mathcal{T}_{\cdot,i} \times \{j\} \subset \hat{\mathcal{D}}. \tag{125}$$

Since $\mathbf{T}$ is invertible, we have
$$\det(\mathbf{T}) = \sum_{\sigma \in \mathcal{S}_{n_A}} \left( \text{sgn}(\sigma) \prod_{j=1}^{n_A} \mathbf{T}_{\sigma(j),j} \right) \neq 0, \tag{126}$$

where $\mathcal{S}_{n_A}$ is a set of $n_A$-permutations. Then there must exist at least one non-zero term in the summation, which indicates that
$$\exists \sigma \in \mathcal{S}_{n_A}, \, \forall j \in \{1, \ldots, n_A\}, \, \text{sgn}(\sigma) \prod_{j=1}^{n_A} \mathbf{T}_{\sigma(j),j} \neq 0. \tag{127}$$

Clearly, there cannot be any term in the product that equals zero, so we have
$$\exists \sigma \in \mathcal{S}_{n_A}, \, \forall j \in \{1, \ldots, n_A\}, \mathbf{T}_{\sigma(j),j} \neq 0. \tag{128}$$

Thus, it follows that
$$\forall i \in \{1, \ldots, n_A\}, \sigma(i) \in \mathcal{T}_{\cdot,i}. \tag{129}$$

Then it yields
$$\forall (i,j) \in \mathcal{D}, (\sigma(i),j) \in \mathcal{T}_{\cdot,i} \times \{j\}. \tag{130}$$

Because of Eq. (125), we have
$$\forall (i,j) \in \mathcal{D}, (\sigma(i),j) \in \hat{\mathcal{D}}. \tag{131}$$

Let us denote $\pi(\mathcal{D})$ as a row permutation of $\mathcal{D}$, where $\forall (i,j) \in \mathcal{D}$, there must be
$$(\sigma(i),j) \in \pi(\mathcal{D}). \tag{132}$$

And it also implies
$$|\pi(\mathcal{D})| = |\mathcal{D}|. \tag{133}$$

Furthermore, Eq. 131 indicates that
$$\pi(\mathcal{D}) \subset \hat{\mathcal{D}}, \tag{134}$$

We have the following relation based on the sparsity regularization:
$$|\hat{\mathcal{D}}| \leq |\mathcal{D}|. \tag{135}$$

Therefore, we have
$$|\pi(\mathcal{D})| = |\mathcal{D}| \geq |\hat{\mathcal{D}}|. \tag{136}$$

Together with Eq. (134), it follows that
$$\hat{\mathcal{D}} = \pi(\mathcal{D}). \tag{137}$$

Thus, we have proved the identifiability of $\mathcal{D}$ up to a permutation on the row indices. Since $M$ is a binary matrix with the support of $\mathcal{D}$, we have proved the connective structure between classes and concepts up to a row permutation. $\square$

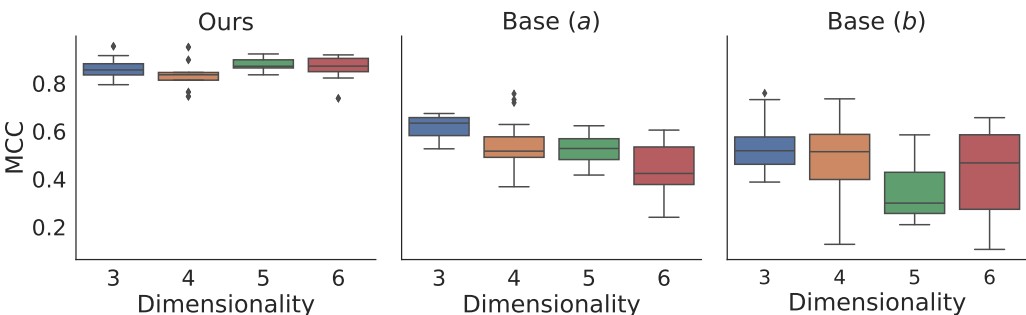

Figure 11: Identification of concepts w.r.t. different numbers of concepts and different settings.

# C. Experiments

In this section, we provide more details regarding the experimental setup as well as additional experimental results to further support our theoretical findings.

## C.1. Supplementary Experimental Setup

We generate the data following the process outlined in our theorems. For our model that identifies only class-dependent concepts (Fig. 5), the connective structure between classes and concepts is generated according to the *Structural Diversity* condition. For class-dependent concepts, we sample from two multivariate Gaussian distributions with zero means and variances drawn from a uniform distribution on $[0.5, 3]$, consistent with parameters used in previous work (Khemakhem et al., 2020b; Sorrenson et al., 2020). For our model that identifies all hidden concepts, including class-independent ones (Fig. 6), the connective structure between class-independent concepts and observed variables follows the structural condition in Prop. 1. These class-independent concepts are sampled from a single multivariate Gaussian distribution with zero means and variances drawn from a uniform distribution on $[0.5, 3]$. In the base model, we remove the structural constraints on both types of connective structures to verify the necessity of the proposed conditions. All other settings remain the same as ours.

In our model evaluation, we employ the Mean Correlation Coefficient (MCC) to measure the alignment between the ground-truth and the recovered latent concepts, which is standard in the literature (Hyvärinen & Morioka, 2016). To calculate MCC, we first compute the pairwise correlation coefficients between the true concepts and the recovered concepts after applying a component-wise transformation via regression. Following this, we solve an assignment to match each recovered concept to the corresponding ground-truth concept with the highest correlation.

We use Generative Flow (Kingma & Dhariwal, 2018) as the nonlinear generating function. For synthetic settings, the sample size is set as $10,000$. Experiments are conducted using the official implementation of GIN[2] (Sorrenson et al., 2020) with an additional $\ell_1$ regularization on the Jacobians and FrEIA[3] (Ardizzone et al., 2018-2022) for the flow-based generative function. The regularization parameters $\lambda$ is set according to a search in $\lambda \in \{0.01, 0.1, 1\}$, and we select $\lambda = 0.1$ according to the average MCCs of experiments conducted on synthetic datasets. Moreover, all experiments are conducted on 12 CPU cores with 16 GB RAM.

## C.2. Supplementary Experimental Results

**Partial violation of previous conditions.** We also conduct experiments to evaluate the identification under partial violations of previously established assumptions in the literature of latent variable models. Specifically, we generated datasets with the following conditions:

1. *Base* $(a)$: The structural sparsity assumption on the mixing structure between latent concepts and observed variables, as outlined in (Zheng et al., 2022), is partially violated for a subset of concepts, with the size randomly selected from all integers in the range 1 to $n/2$.

---

[2]https://github.com/VLL-HD/GIN
[3]https://github.com/vislearn/FrEIA

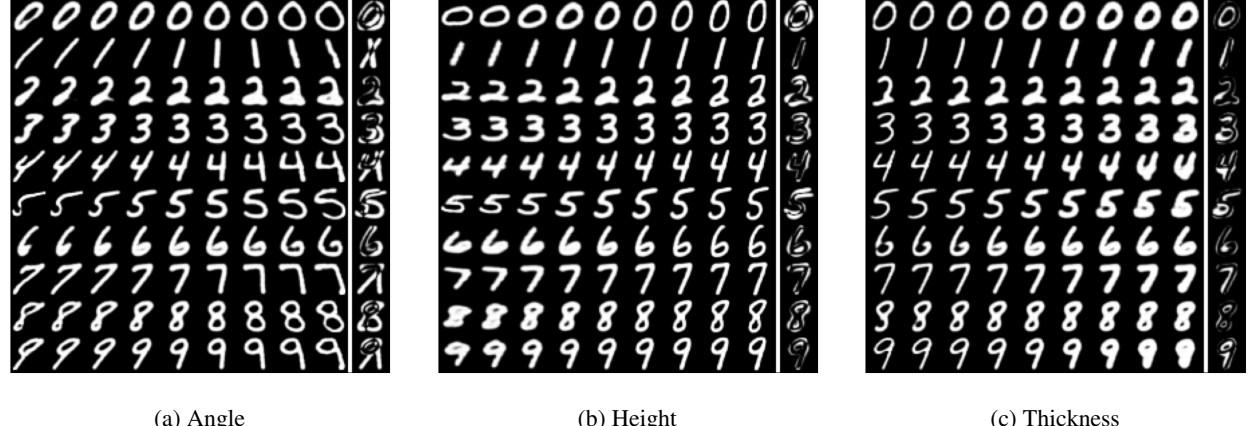

| (a) Angle | (b) Height | (c) Thickness |

Figure 12: Results for each digit class in the EMNIST dataset, showing the identified concepts with the top three standard deviations (SDs). Each subfigure represents a concept identified by our model, with values ranging from $-4$ to $+4$ SDs to demonstrate their impact. The rightmost column features a heat map of the absolute pixel differences between $-1$ and $+1$ SDs. These concepts can be interpreted as variations in angle, height, and thickness.

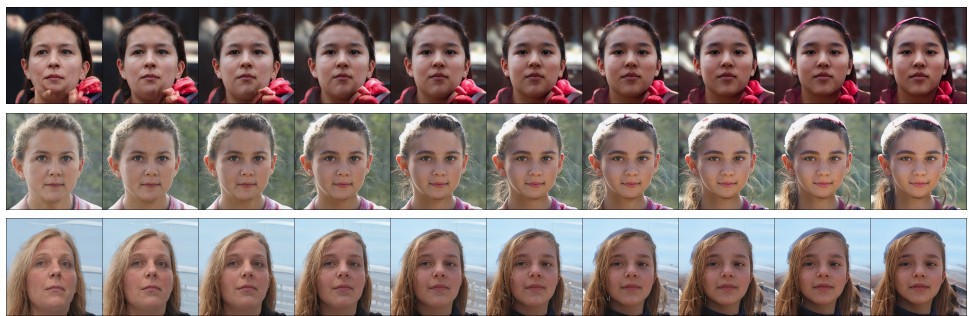

Figure 13: Multiple concepts (e.g., skin, eyes, face shape, etc.) corresponding to "Age" are entangled.

2. *Base* ($b$): The $2n+1$ domain requirement in (Khemakhem et al., 2020b; Kong et al., 2022) is partially violated. Instead, latent concepts are generated from $n+1$ multivariate Gaussian distributions, each with zero mean and variances drawn from a uniform distribution over $[0.5, 3]$.

3. *Ours*: The data-generating process adheres to our proposed structural diversity condition. While there are no constraints on the mixing structure between latent concepts and observed variables, the structure between classes and concepts satisfies the required structural diversity.

The results, shown in Fig. 11, indicate that when assumptions from previous works are partially violated, the recovery of latent concepts becomes unreliable, shedding light on the necessity of the proposed flexible guarantees based on learning by comparison. All results are from 10 runs with different random seeds.

**Some concepts are naturally entangled.** As discussed in Sec. 4, we include the results on EMNIST and FFHQ datasets here in the appendix. The EMNIST dataset (Cohen et al., 2017) is an extension of the classical MNIST, which consists of a much larger set of handwritten digits derived from the NIST Special Database 19 (Grother & Hanaoka, 1995).

The results are shown in Fig. 12. Similar to the other datasets, we select the identified components with the top three standard deviations and vary the value of the identified components to visualize their potential semantics. According to the results, it is clear that the hidden concepts can be identified by only learning from diverse classes of observations. This further indicates that the proposed nonparametric identifiability, which is based on the basic cognitive mechanism of learning by comparison, has potential applicability in real-world scenarios.

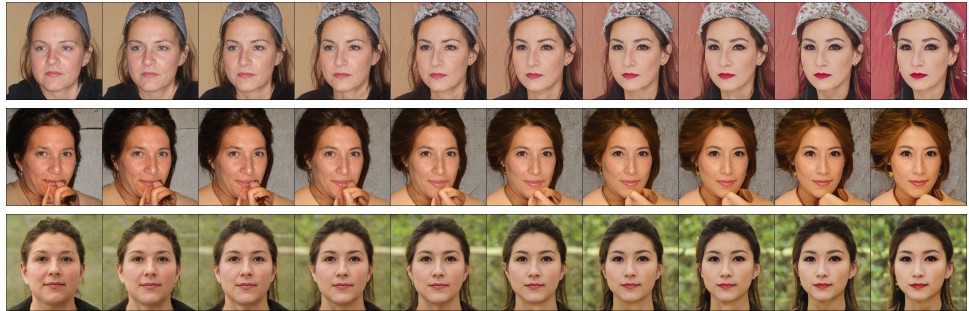

Figure 14: Multiple concepts (e.g., lipstick, eye shadow, powder, etc.) corresponding to "Makeup" are entangled.

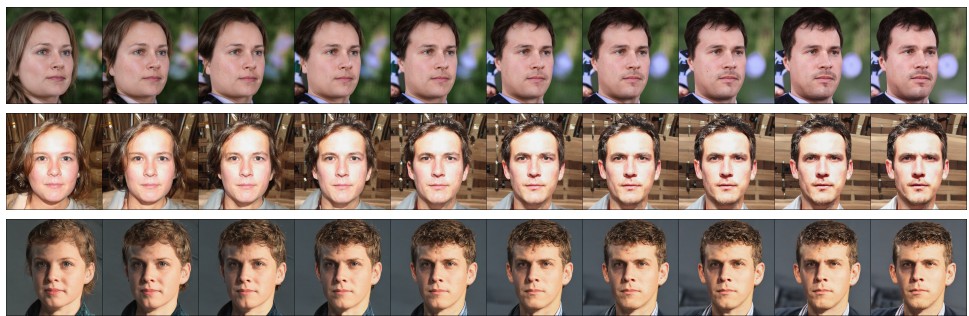

Figure 15: Multiple concepts (e.g., hairstyle, head shape, eye, etc.) corresponding to "Gender" are entangled.

To explore scenarios where not all concepts can be identified element-wise, we conduct additional real-world experiments on a more complex scenario, i.e., the FFHQ dataset (Karras et al., 2019). The dataset contains $70,000$ human face images, which is more complicated than the datasets in our other experiments.

From Figs. 13, 14, and 15, it is evident that some concepts remain entangled and cannot be fully recovered. For instance, for the class "Age", concepts like "skin," "eye," and "face shape" are all entangled together, suggesting that assumptions for component-wise identifiability may not be fully satisfied in this scenario. However, these class-dependent concepts can still be identified as a group, consistent with our theorem based on local or pairwise comparisons. This suggests that, even in complex scenarios where theories fail to guarantee identifiability for all individual concepts due to assumption violations, our alternative identifiability framework based on pairwise comparisons may still provide an alternative theoretical basis for recovering class-dependent concepts collectively, even if they remain entangled. This sheds light on the necessity of our alternative identifiability guarantees in some complicated real-world scenarios.

**Learning concepts with noise.** We evaluate the robustness of concept recovery in noisy environments. Specifically, we introduce non-additive random noise to the Fashion-MNIST images, since additive noise has been extensively studied and can often be removed via deconvolution. The more challenging case with non-additive noise is also shown to be identifiable with rather general conditions (Zheng et al., 2025). Example noisy samples are shown in Fig. 16. Despite the corruption, our method still recovers meaningful concepts (Fig. 17), demonstrating the framework's robustness to general noise.

**Quantitative evaluation on images.** To quantitatively evaluate our framework on images, we conducted a series of experiments. Specifically, we report Mutual Information Gap (MIG) (Chen et al., 2018) and DCI scores (Eastwood & Williams, 2018) for synthetic datasets (Figures 18 and 19), and Fréchet Inception Distance (FID) (Heusel et al., 2017) and Perceptual Path Length (PPL) (Karras et al., 2019) for real-world datasets, CUB-200-2011 (Wah et al., 2011) and AWA2 (Xian et al., 2018). CUB-200-2011 contains 200 bird classes with 11,788 images; AWA2 includes 50 animal categories with 37,322 images. We report a Mutual Information Gap (MIG) of $8.135$, a DCI score of $16.533$, a Fréchet Inception Distance (FID) of $8.11$, and a Perceptual Path Length (PPL) of $30.938$, indicating strong disentanglement and high-quality generation under our framework. We also visualize the recovered concepts from CUB-200-2011 and AWA2 (Figs. 20 and 21).

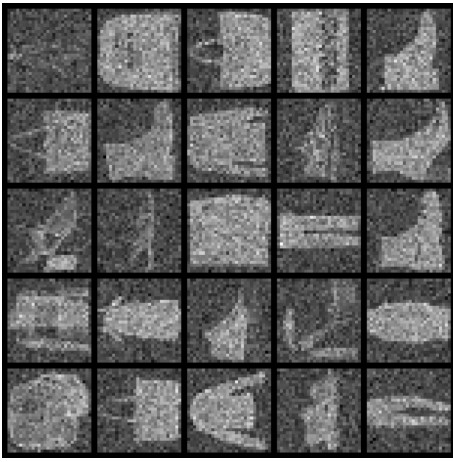

Figure 16: Samples of Fashion-MNIST with noise.

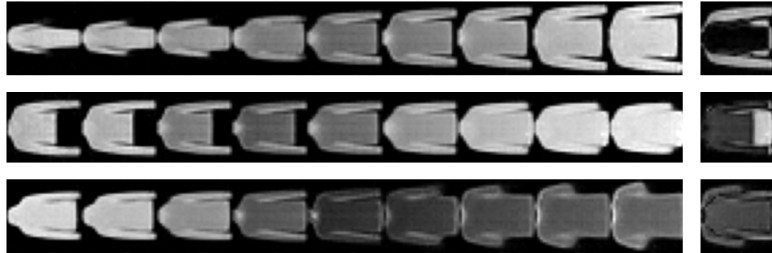

Figure 17: Results on *noisy* Fashion-MNIST. The rows correspond to different identified concepts: "sleeve length," "torso length," and "shoulder width," respectively.

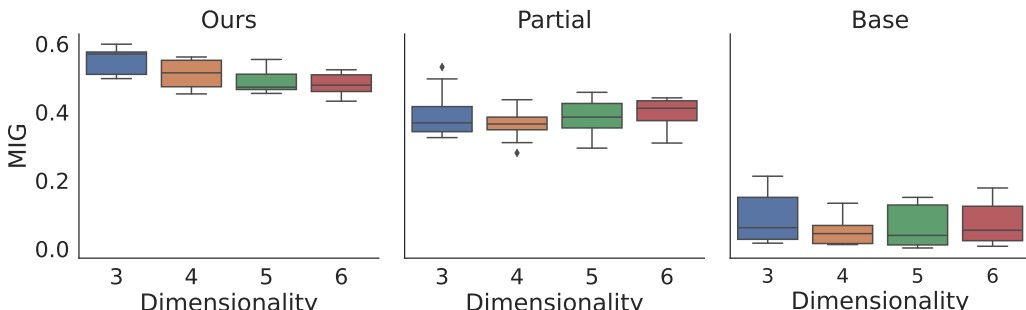

Figure 18: Mutual Information Gap (MIG) w.r.t. $10,000$ samples and different number of concepts. All results are from 10 runs with different random seeds. The settings for *Ours* and *Base* follow those used in the main manuscript. For *Partial*, we intentionally violate the structural diversity assumption for one randomly selected concept and vary the total number of concepts to control the proportion of the violated concept.

## D. Supplementary Discussion

### D.1. Challenges, Motivations, and Implications

**Challenges of nonparametric identifiability.** Theoretical and empirical evidence consistently demonstrates that, without additional assumptions, achieving identifiability in nonparametric generative processes is nearly impossible (Hyvärinen & Pajunen, 1999; Locatello et al., 2019). This challenge extends beyond concept learning to fields such as independent component analysis and causal representation learning. Without additional conditions, it is impossible to ensure that the recovered concepts align with the ground truth, as the solution space is vastly underconstrained, allowing non-trivial mixtures

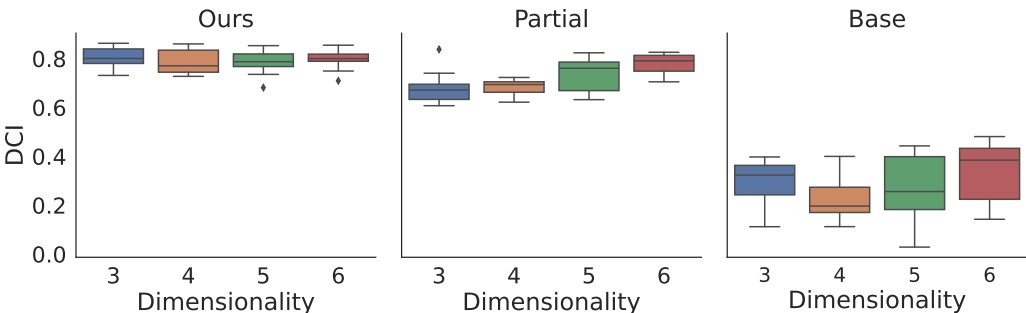

Figure 19: Disentanglement, Completeness, and Informativeness (DCI) Disentanglement score w.r.t. $10,000$ samples and different number of concepts. All results are from 10 runs with different random seeds. The settings for *Ours* and *Base* follow those used in the main manuscript. For *Partial*, we intentionally violate the structural diversity assumption for one randomly selected concept and vary the total number of concepts to control the proportion of the violated concept.

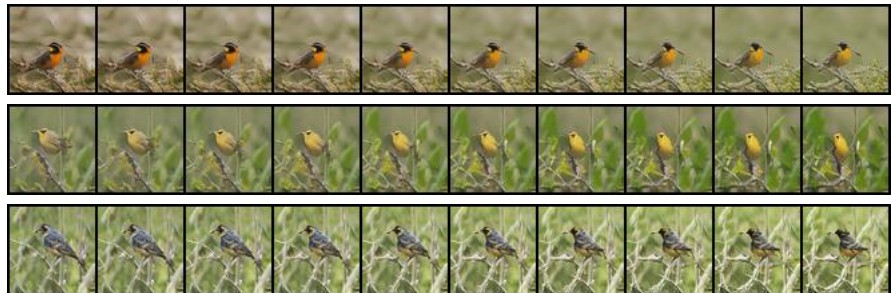

Figure 20: Results on CUB-200-2011. The rows correspond to different identified concepts: "color," "orientation," and "size," respectively.

of the true concepts to generate observationally equivalent data. Parametric assumptions, such as linearity, disjointness, or additivity, constrain the solution space, reducing ambiguity and identifying the true generative process up to certain indeterminacies. In contrast, our focus is on understanding what concepts can be recovered in the general setting. Our nonparametric identifiability results provide insights into this general setting and suggest a promising direction.

**Learning by pairwise comparison.** Theorem 1 demonstrates that for any given pair of classes and their corresponding sets of hidden concepts, the unique concepts in each class can be disentangled from all the remaining concepts. This process is fundamental to the cognitive mechanism of learning through comparison. Consider an infant with no prior experience of the world: when presented with two classes, such as a cat and a dog, the infant learns and memorizes the unique concepts associated with each class, such as "meows" for the cat and "barks" for the dog. The invariant concepts, like "furry" or "four-legged," cannot be distinctly learned because they do not provide distinguishing information between the classes. From a cognitive science perspective, infants and young learners rely heavily on contrastive features to form distinct categories and concepts (Eimas et al., 1971). For instance, if an infant repeatedly hears a cat meow and a dog bark, they begin to associate these unique sounds with the respective animals. In contrast, shared attributes like fur or four legs do not stand out because they do not help in differentiating between the two animals. This emphasizes the role of unique concepts in early learning and memory, highlighting how pair-wise comparisons are essential in the process of discovering the new world. For machines to learn without prior knowledge, we argue that similar mechanisms also help.

**Learning by local comparison.** Corollary 1 extends these theoretical guarantees from pair-wise comparisons to local comparisons among multiple classes. Although pairwise comparison is fundamental to the learning mechanism, local comparison is more efficient in complex scenarios. For instance, when an infant is exposed to a variety of stimuli, they do not learn by isolating pairs indefinitely. Instead, they begin to discern patterns and unique features within a broader context, comparing multiple classes simultaneously. For example, a child distinguishing between a cat, a dog, and a bird must identify unique concepts such as "meows," "barks," and "chirp." As the child interacts with these animals in different contexts—perhaps hearing a bird chirp in the park, a dog bark at home, and a cat meow in the neighbor's yard—they learn to

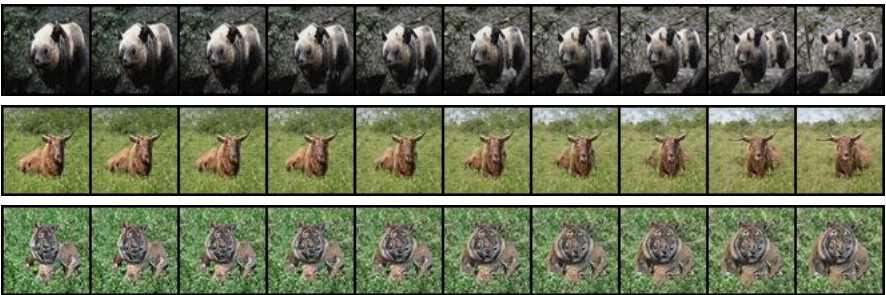

Figure 21: Results on AWA2. The rows correspond to different identified concepts: "number of animals," "orientation," and "size," respectively.

associate specific sounds and behaviors with each animal. This local comparison ensures that even as the number of classes increases, the child can efficiently disentangle and learn the unique concepts of each class, providing a more complete understanding of the new environment.

**Class-concept alignment.** Proposition 2 indicates that, the recovered hidden structure between classes and concepts is an isomorphism of the ground-truth structure. Intuitively, this helps the machine understand which concepts correspond to a given class of observations. While this process may seem straightforward to us, it can be challenging for infants or machines without prior experience, as it aligns with an essential step of learning through comparison. For instance, consider an infant presented with a set of objects like a cat, a dog, and a bird (the classes) and a set of concepts like "furry," "barks," and "flies." Without proper knowledge, the infant might incorrectly assign "barks" to the cat or "flies" to the dog, lacking the experience to accurately match these concepts with the correct classes. The concept of "furry" might also be mistakenly assigned to the bird, despite its inapplicability. Therefore, to distinguish different classes by their concepts and learn unique concepts through comparison, the machine must first recover the underlying connective structure. This is essential for provably learning from multiple classes of observations.

### D.2. Supplementary Examples for Assumptions

We first include a concrete example for assumptions in Thm. 1 and Cor. 1 as follows:

*Example* 5. Suppose there exist two samples with their corresponding Jacobians given by $D_{\mathbf{c}}g((c,\theta)^{(1)})_{:,i} = (0,1,2)$ and $D_{\mathbf{c}}g((c,\theta)^{(2)})_{:,i} = (0,3,4)$. Clearly, these two vectors span a 2-dimensional subspace. We can also find a matrix T (e.g., a binary matrix with the same support as **T**) s.t. $\left[\mathrm{T}D_{\mathbf{c}}g((c,\theta)^{(\ell)})\right]_{:,i} \in \mathbb{R}^{n_A}_{\hat{\mathcal{D}}_{:,i}}$ for $\ell \in \{1,2\}$. Since identifiability theory considers an infinite number of samples, the requirement for several non-degenerate samples is almost always satisfied asymptotically.

Then we introduce an illustrative example for the distributional variability condition in Thm. 2. Intuitively, the condition is a generic faithfulness assumption, ruling out special parameter combinations that would make the two integrals equal.

*Example* 6. Consider **c** as a 2-dimensional vector with $c^{(k)} = [1,0]$ and $c^{(v)} = [0,1]$. Let $\mathcal{Z} = \mathbb{R}^2$, and $A_{\mathbf{z}} = \{(z_1, z_2) \in \mathbb{R}^2 : 0 \le z_1 \le 2, 0 \le z_2 \le 1\}$. The conditional densities are $p(\mathbf{z} \mid \mathbf{c} = [1,0]) = \frac{1}{2\pi}e^{-\frac{(z_1-1)^2+(z_2-0)^2}{2}}$ and $p(\mathbf{z} \mid \mathbf{c} = [0,1]) = \frac{1}{2\pi}e^{-\frac{(z_1-0)^2+(z_2-1)^2}{2}}$. Evaluating the integrals over $A_{\mathbf{z}}$, we have

$$\int_0^1 \int_0^1 \frac{1}{2\pi}e^{-\frac{(z_1-1)^2+(z_2-0)^2}{2}} dz_1 dz_2 \neq \int_0^1 \int_0^1 \frac{1}{2\pi}e^{-\frac{(z_1-0)^2+(z_2-1)^2}{2}} dz_1 dz_2.$$

Note that $(k,v)$ can even be different for different $A_{\mathbf{z}}$, which further weakens the assumption.

