# OpenReview forum: "Nonparametric Identification of Latent Concepts"
_ICML.cc/2025/Conference — ICML 2025 poster_

### Official Review · Reviewer_CTeq · 2025-02-24

**Overall Recommendation:** 3

**Summary:**

The paper proposes that when there is an unknown concept-to-class matrix, the concepts can be identified under certain assumptions on the structure of the matrix. Namely, it formalizes in which sense each concept needs to have characteristic classes it belongs to, or vice versa, how every class needs to have characteristic attributes that distinguish it from the other classes. The paper is primarily theoretical, with some synthetical experiments.

## Update after the rebuttal

I am raising my score to 3 because $f$ is non-linear (though still bijective), $z$ is continuous, and because of the inclusion of DCI/MIG, FID/PPL, and CUB/AWA experiments.

**Claims And Evidence:**

Under the given assumptions, concepts are identifiable

* This seems to be theoretically backed, but I did not check the proofs

The theoretical identifiability holds in practice

* I have doubts here since the experimental setup is very small-scoped, see below.

**Essential References Not Discussed:**

The paper discusses most of the relevant works in nonlinear ICA.

**Experimental Designs Or Analyses:**

Synthetical experiments lack metrics, models, and baseline approaches.

The paper lacks quantitative real-life dataset experiments.

**Methods And Evaluation Criteria:**

The paper uses only one metric to benchmark whether the captured concepts are ground-truth concepts, the ECC. While widely used, there are other popular metrics like the Disentanglement-Completeness-Informativeness score (DCI) or the Separated Attribute Predictability (SAP). It would be more convincing to test on multiple criteria, since they are known to all be noisy and imperfect.

The paper does not compare to any baselines, except the base model.

All quantiative experiments are only on synthetical datasets. Real datasets only have qualitative plots, without baselines.

**Other Comments Or Suggestions:**

I did not find typos or formatting issues.

**Other Strengths And Weaknesses:**

Strengths

* Bridging concept learning literature with nonlinear ICA / identifiability literature is overdue
* There are no distributional assumptions on the latent concepts, in particular no non-Gaussianity (though I am not sure, because the paper seems to exclude fully Gaussian data)


Weaknesses

* There are very strong assumptions / the space of generative processes this can be used for is very reduced
    * There is a very strong assumption that the generative process $f$ is bijective.
    * It may be that $f$ is linear (if it denotes M), though I am not sure, because the spaces that f works in are not defined (class to concept, class to image, image to concept, ...) and no comment on linearity / non-linearity is made
    * All concepts are binary
    * There is the structural diversity assumption, and I cannot judge how strong it is. It assumes that for each concept, there exists at least one class that has it (so far so good) and there exists a set of classes that does not have it, and this concept is the only concept that is not found in any of these classes. I understand that this prevents block matrices (which are by default non-identifiable) but cannot judge how many other realistic matrices it assumes away.
* The experiments are on only one model, with one evaluation metric, and only on synthetical datasets (all evaluations on the real-life datasets are purely qualitative and without a baseline). It can thus be very overfitted to that setup, since, as put in the appendix “We generate the data following the process outlined in our theorems.”. Showing that the approach works on a synthetical setup which exactly follows its the paper’s assumptions is not a very general result, especially when the assumptions are as strict as in this paper.
* The experiments do not compare to any baseline approaches, except the base model, see also below.
* Code is not provided.

Score justification:
The paper seems to make progress on the exact matrix structure that is needed to allow identifiability, but I cannot quite judge how much it differs from current approaches since I know enough about ICA to not consider myself an expert. Hence, in terms of novelty and theoretical contribution, the score can be anywhere between 2-4. I am currently more on the sceptic end of this range for two reasons: 1) I believe the assumptions are rather strong / the applicability is limited to binary matrices (since f needs to be bijective and M binary). This is understandable for structure learning, but limiting for concept learning. I hope that the other reviewers can comment on the strength of the assumptions and novelty over previous approaches. 2) the experiments are very limited, to one evaluation metric and one model and only on synthetical datasets that exactly follow the assumptions the approach exploits. This limits the generality of the results. More results, as outlined above and below, can convince me otherwise here.

Post rebuttal score: I have raise to score 3 due to the additional experiments and due to clarifications on the assumptions, which are still restrictive, but not as restrictive as I initially believed.

**Questions For Authors:**

If I understand correctly, your approach should work quite well on attribute-based classification, such as on CUB or AwA2? There you have a (binarizable) attribute-per-class matrix. Can you comment on whether it applies, and test your algorithm on it? This would help judge if it works in higher dimensions.

I am concerned about how far the results span in terms of width of generative processes covered. Could you compare to the following two neighboring approaches (https://openreview.net/pdf?id=mGUJMqjDwE, https://proceedings.mlr.press/v216/leemann23a/leemann23a.pdf) in structural identifiability in terms of...

* Theoretical scope, i.e., the assumptions placed on the generative process to guarantee identifiability
* Practical applicability, i.e., which datasets it can currently discover concepts on

**Relation To Broader Scientific Literature:**

The authors make connections between concept learning, nonlinear ICA, and structure learning.

**Theoretical Claims:**

I do not consider myself an expert in the very nuanced proofs of identifiability / nonlinear ICA, so I abstain from guaranteeing correctness of the proofs.

---

> ### Author Rebuttal · Authors · 2025-04-01
>
> We are genuinely grateful for your insightful comments. In light of these, we have introduced **new experiments** (https://anonymous.4open.science/r/0-518C/rebuttal_new_results.pdf) and new discussions. Please find our detailed response below:
>
> **Q1:** Strong assumptions / limited generative process space.
>
> **A1:** Thank you so much for the comment. It seems we may not be fully aligned on the core setting and assumptions. We sincerely appreciate the opportunity to clarify the following:
>
> - **Q1.1:** There is a very strong assumption that the generative process $f$ is bijective.
>
>   **A1.1:** As defined in L129-136 left, $f$ is an unknown diffeomorphism onto its image—a very *standard* assumption in identifiability literature (e.g., most works mentioned in the survey [Hyvärinen et al. 2024]). It aligns with the manifold hypothesis, which posits that high-dimensional data (e.g., pixels) lie on a lower-dimensional latent manifold (e.g., concepts). In overcomplete settings (latent dim > observed dim), much stronger assumptions (e.g., linearity) are typically required.
>
>   - *Hyvärinen et al., Identifiability of latent-variable and structural-equation models: from linear to nonlinear, 2024*
>
> - **Q1.2:** $f$ may be linear (if it denotes $M$); unclear spaces that $f$ maps between.
>
>   **A1.2:** Thanks for raising this. To clarify, **$f$ and $M$ are unrelated**.
>
>    - As defined in L113–121 left, $f$ maps latent concepts $\mathbf{z}$ to observations $\mathbf{x}$ (Eq. 1: $\mathbf{x} = f(\mathbf{z})$) and is a general diffeomorphism, **without any linearity constraint**.
>
>    - As defined in L159 left–133 right, $M$ is a binary structural mask between classes $\mathbf{c}$ and concepts $\mathbf{z}$, independent of $f$.
>
> - **Q1.3:** All concepts are binary.
>
>   **A1.3:** As defined in L116 left, all concepts are **continuous variables** ($\mathbf{z} \subseteq \mathbb{R}^{n}$), with no restriction to binary values.
>
> - **Q1.4:** Unclear how strong the structural diversity is.
>
>   **A1.4:** Thank you for the great question. In a general nonparametric setting (*no functional/distributional constraints*), identifiability is known to be ill-posed without additional assumptions. Structural diversity characterizes when *every concept* is identifiable in such settings. It is expected that ambiguous cases must be excluded, as in most prior identifiability theories.
>
>   A key contribution of our work is going *beyond* this assumption. The local comparison framework (Thm. 1, Prop. 1) provides identifiability guarantees even when global condition is *violated*—allowing recovery of many concepts as groups (e.g., in FFHQ). In contrast, most prior work cannot provide any guarantees under any degree of violation.
>
>   In light of your question, we have also added new experiments under partial violations (Figs. 1–2 in the link), confirming empirical robustness.
>
> **Q2:** Additional metrics for synthetic and real-world datasets.
>
> **A2:** Great suggestion. In light of it, we have added evaluations on DCI and Mutual Information Gap (MIG) for synthetic datasets (Figs. 1–2 in the link), and Fréchet Inception Distance (FID) and Perceptual Path Length (PPL) for real-world images (Table 1 in the link). Our method consistently outperforms baselines.
>
> **Q3:** More baselines.
>
> **A3:** Thanks for your advice. We have added two more baselines (Table 1 and Figs. 7-8 in the link), further supporting the effectiveness.
>
> **Q4:** Code is not provided.
>
> **A4:** Thanks. Due to rebuttal policy, we can only include figures in the link. Based on the flow-based method (L1372), we added an ℓ₁ norm (L369 right) since ℓ₀ norm is known to be indifferentiable.
>
> **Q5:** Test on CUB or AwA2.
>
> **A5:** Thank you for the advice. We have conducted new experiments on both datasets (Figs. 5–8, Table 1 in the link), showing that identifiable concepts remain recoverable. With these additions, we now have *seven* different real-world datasets to support the practical applicability.
>
> **Q6:** Comparison with two related papers.
>
> **A6:** Thanks for the suggestion. Please find the detailed comparison below:
>
> - **Brady et al., 2023:**
>
>    (This work has been discussed in the introduction (L62-76 left).)
>
>   - ***Theory:*** Assumes each observed variable (pixel) maps to only one latent slot.
> 	- In contrast, we allow arbitrary mixing.
>
>
>   - ***Practice:*** Targets recovering concrete objects as pixel groups, assuming no occlusion or overlap among them.
> 	- In contrast, we focus on general concepts (concrete or abstract), with no constraints on observed object composition.
>
>
> - **Leemann et al., 2023:**
>
>   - ***Theory:*** Assumes linear generative process or orthogonal Jacobians.
> 	- In contrast, we allow general nonlinear generative process, where the identifiability is much harder.
>
>   - ***Practice:*** Targets post-hoc explanation assuming access to linear combinations of latent factors.
> 	- In contrast, we focus on general concept learning from observation in a nonparametric setting.

---

> > ### Comment · Reviewer_CTeq · 2025-04-03
> >
> > Thank you for clarifying the points. I am raising my score to 3 because $f$ is non-linear (though still bijective), $z$ is continuous, and because of the inclusion of DCI/MIG, FID/PPL, and CUB/AWA experiments.

---

> > > ### Author Response · Authors · 2025-04-03
> > >
> > > Thank you very much for your kind encouragement! We're delighted that your concerns have been addressed. We deeply appreciate your time, thoughtful feedback, and valuable suggestions, which have greatly helped us improve our manuscript.

---

### Official Review · Reviewer_kdtg · 2025-03-08

**Overall Recommendation:** 4

**Summary:**

The paper studies the identifiability of latent concepts. The key assumption is that we observe class labels $c$ and observations $x$ which are mediated through concepts $z$, i.e., $c\to z\to x$ and they in addition allow for class independent features. Then they show that the difference in concepts for different classes can be identified and under a diversity condition the class dependent features are identifiable and the remaining features are block identifiable. Then they conduct experiments on synthetic and real data using established methods to investigate their identifiability results.

## update after rebuttal

As explained in the rebuttal comment,  I maintain my (overall positive) opinion

**Claims And Evidence:**

The key contributions of the paper are theoretical, see below.

**Essential References Not Discussed:**

One could consider citing more works from the causal representation learning literature that relies on similar techniques, but the key original works seem to be cited.

**Experimental Designs Or Analyses:**

My understanding is that the authors do the following:
They use (up to minor changes) a method from prior work and the comparison is not to another method but to a setting where their assumptions are violated. Is this correct? In this case (especially the second part) should be made a bit clearer in the text.

**Methods And Evaluation Criteria:**

Yes, the datasets make sense, for evaluation see below.

**Other Comments Or Suggestions:**

l 117 left: There seems to be something missing.

The set $A$ and the matrix $M$ should be related explicitly

$\theta$ in Definition 1 is a abuse of notation because $\theta$ was used before

l.197 Definition of $T$ is unclear. Do you define $\mathbf{T}$ by the equation that follows? Make this clearer

Proposition 1 might be renamed to Corollary 1

l 266 left: This is hard to understand, do you mean 'and which cannot be expressed as'? Do you allow singular distributions on $z_A$?

l 360 right: How exactly are the $c^{(i)}$ created? What is a multi-hot vector? Is is sufficient to use one-hot vectors?

Learning by comparison could be connected closer to the experimental methodology and the literature

**Other Strengths And Weaknesses:**

Strengths:

The paper is well motivated and overall well written. The examples are well-chosen, and it combines a setting of relevance (concept learning) with rigorous theoretical results that rely on reasonable assumptions. In particular, the results do not rely on parametric assumptions which makes them quite general. The discovered concepts in the experiments are quite convincing.

Weakness:

The technical novelty is limited. The experimental methodology is not new. The results require quite some supervision (class labels) this is a weakness compared to some existing work.

**Questions For Authors:**

See theoretical claims and experimental design. Depending on the answer to the former my assessment would change.

**Relation To Broader Scientific Literature:**

The specific setup of the paper is novel to my knowledge and I find it quite convincing, essentially it seems to be, loosely spearking, a combination of the iVAE paper with structural sparsity. The theoretical results are also new, but the used techniques date back to the first works on identifiability of linear ICA and the iVAE work and the sparsity approaches have also similarly appeared before.

**Theoretical Claims:**

The results generally seem plausible. However, I have one question/complaint. The treatment of the $\theta$ variable is a bit unclear. So the initial relation $(4)$ is clear. But then a relation for $g$ is used in (5) but $g$ is not injective and depends on $\theta$. In my opinion this part needs to be clarified. I did not check the rest of the argument in detail but it seems to follow the standard arguments.

---

> ### Author Rebuttal · Authors · 2025-04-01
>
> We deeply appreciate your valuable insights. In light of these, we have carefully added several new discussions and conducted **new experiments** (https://anonymous.4open.science/r/0-518C/rebuttal_new_results.pdf). Please find our detailed responses below:
>
> **Q1:** More clarifications on Eq. 5.
>
> **A1:** Thanks for your suggestion, we have added further clarifications accordingly. $\mathbf{T}$ in Eq. 5 is invertible because the mapping $t$ is invertible. The invertibility of $t$ follows from Eq. 4 (L743-746), independent of $g$. Eq. 5 simply represents the derivatives w.r.t. $\mathbf{c}$, and $\theta$ denotes other factors independent of $\mathbf{c}$.
>
>
> **Q2:** Is the comparison based on the same method under violated assumptions?
>
> **A2:** Yes. The estimation method (previous work + regularization) is unchanged; only the data-generating process is altered to violate structural conditions (L407 left). We have now emphasized this more explicitly in light of your great question.
>
> **Q3:** The technical relation to previous methods such as iVAE and sparsity approaches.
>
> **A3:** Thanks a lot for the insightful comment. iVAE (and related methods) require sufficient distributional change (e.g., $2n+1$ environments), while we focus on how classes affect concepts—structure, not magnitude, of change. Our theory is closer in spirit to sparsity-based methods, as both leverage structural relationships. However, there are two key differences:
> - We study class–concept structures, while prior work focuses on concept–measurement structures. This relaxes assumptions on mixing function by weak supervision from class labels.
>
> - Our pairwise comparison framework supports identifiability under partial violations, unlike prior work that requires full structural assumptions.
>
> We hope this makes the distinctions clearer and would be more than happy to further refine it based on your feedback.
>
> **Q4:** Possible missingness in L117 left.
>
> **A4:** Thanks. We were not certain we understood correctly, so we assumed you suggested moving L124-125 ($p(\mathbf{z}|\mathbf{c}) = p(\mathbf{z}_A|\mathbf{c}) p(\mathbf{z}_B)$,  $\mathbf{z}_A \coloneqq g(\mathbf{c}, \theta)$) alongside L117 (Eq. 1). We've made this change, but please let us know if this wasn't your intention.
>
>
> **Q5:** The set $A$ and the matrix $M$ should be related explicitly.
>
> **A5:** Thank you–we now explictly highlight that $A_i$ is the support of $M_{i,\cdot}$ in the updated manuscript.
>
> **Q6:** Abuse of notation with $\theta$ in Defn. 1.
>
> **A6:** Addressed—we removed $\theta$ in Defn. 1, keeping only $(g, p_{\mathbf{z}}, M)$, and likewise for $\theta’$. Thanks for raising this.
>
> **Q7:** More clarification on $\mathbf{T}$ and $\mathrm{T}$.
>
> **A7:** Thank you for your suggestion. We have added further highlights as follows:
>
> - $\mathbf{T}$: a matrix-valued function in $D_{\mathbf{c}} \hat{g} = \mathbf{T} D_\mathbf{c} g$.
>
> - $\mathrm{T}$: a matrix with the same support as $\mathbf{T}$.
>
> **Q8:** Rename Prop. 1 to Cor. 1.
>
> **A8:** Thanks, we have renamed it accordingly.
>
> **Q9:** L266 left — Do you mean “and which cannot be expressed as”? Do you allow singular distributions?
>
> **A9:** Thanks for the suggestion. We meant “and which cannot be expressed as” and have revised the wording. The condition excludes singular distributions by requiring the subset $A_{\mathbf{z}} \subseteq \mathcal{Z}$ to have non-zero probability measure.
>
> **Q10:** L360 right — How exactly are the $\mathbf{c}^{(i)}$ created? What is a multi-hot vector? Is it sufficient to use one-hot vectors?
>
> **A10:** Thank you for the great suggestions. Accordingly, we have added a detailed explanation in the updated manuscript. Specifically, $\mathbf{c}^{(i)}$ denotes the classes of sample $i$: a one-hot vector if single-labeled, or a multi-hot vector if multi-labeled (e.g., $[0,1,1]$ for two active classes).
>
> **Q11:** Learning by comparison could be connected closer to the experimental methodology and the literature
>
> **A11:** Thanks for the helpful advice. We have rewritten L82–99 left for closer grounding:
>
> > We address this question by grounding our approach in a fundamental cognitive mechanism: humans learn concepts by contrasting diverse classes of observations. Classic studies have shown that concept formation relies on detecting distinctions across examples—Bruner et al. (1956) emphasized learning through contrasts between exemplars and non-exemplars; Gibson (1963, 1969) proposed differentiation as a basis of perceptual learning in infants; and Gentner & Namy (1999) demonstrated that direct comparison enables children to abstract category-defining features. This principle is further supported by extensive literature across cognitive science, reinforcing that learning by comparison is the key underlying engine.
>
> Experimentally, FFHQ (Figs. 13–15) illustrates the need for local comparisons due to entangled concepts. We also added new experiments on two new datasets (see anonymous link), in light of your great insights.

---

> > ### Comment · Reviewer_kdtg · 2025-04-05
> >
> > - There is no need to praise any minimal suggestion.
> > - There was a typo in the review: There is a typo in line 147 left (at least missing bracket), not in 117, please apologize.
> > - After reading the rebuttal and the other reviews, I still think that this is overall a solid contribution.

---

> > > ### Author Response · Authors · 2025-04-05
> > >
> > > Thank you for the clarification—indeed, L147 left should read $\operatorname{supp}(\mathbf{S}) \coloneqq \\{(i,j) \mid \exists \theta \in \Theta, \mathbf{S}(\theta)_{i,j} \neq 0 \\}$. We appreciate your feedback and are glad you consider the contribution solid.

---

### Official Review · Reviewer_VdCS · 2025-03-11

**Overall Recommendation:** 4

**Summary:**

This paper proposes a nonparametric framework for identifying latent concepts by leveraging structural diversity across observation classes, inspired by human cognitive mechanisms of learning through comparison. The authors establish theoretical guarantees for concept identifiability without relying on parametric assumptions about concept types, functional relationships, or generative models. Key contributions include:

1. **Pairwise and local comparison theorems** (Thm. 1, Prop. 1) showing that unique concepts in classes can be disentangled when sufficient diversity exists.
2. **Global identifiability** (Thm. 2) under structural diversity conditions, recovering class-dependent concepts up to permutation/element-wise transforms.
3. **Structure recovery** (Prop. 3) of class-concept relationships.
4. Empirical validation on synthetic data and real-world datasets (Fashion-MNIST, AnimalFace, FFHQ), demonstrating alignment between identified concepts and semantic attributes.


## update after rebuttal

Thanks for the authors' rebuttal which addressed my concern. So I increase my score to clear accept.

**Claims And Evidence:**

The proofs for Thm. 1 (pairwise disentanglement) and Thm. 2 (global identifiability) were examined.

While the linear algebra arguments and Jacobian-based reasoning are logically structured, two concerns arise:

- **Thm. 1** assumes linear independence of Jacobian vectors (Eq. 6). The proof does not address whether this holds in high-dimensional spaces or under noisy observations, potentially limiting practical applicability.
- **Thm. 2** relies on the structural diversity assumption (Assump. 1), which requires each concept to be uniquely tied to a class subset. The proof does not quantify how often this condition fails in real-world scenarios (e.g., overlapping concepts like "furry" for cats/dogs).

**Essential References Not Discussed:**

No essential references are not discussed.

**Experimental Designs Or Analyses:**

Experiments on synthetic data validate identifiability under controlled structural conditions. However:

- Real-world results (e.g., FFHQ in Fig. 13–15) show entangled concepts (e.g., "age" involving skin/eyes), yet the paper claims partial identifiability via local comparisons. This discrepancy is not rigorously analyzed.
- Baseline comparisons (e.g., "Base" models) lack details on architecture/optimization parity, raising questions about fairness.
- The MCC metric measures alignment but does not assess semantic interpretability, a critical aspect of concept learning.

**Methods And Evaluation Criteria:**

**Methods**: The nonparametric framework is novel and well-motivated by cognitive mechanisms. However, the paper does not provide an explicit algorithm for estimating concepts (e.g., how to enforce ℓ0ℓ0 regularization in practice). This limits reproducibility.

**Evaluation**:

- **Synthetic experiments** validate identifiability under ideal conditions but do not test robustness to structural assumption violations (e.g., partial failures of Assump. 1).
- **Real-world evaluations** rely on qualitative interpretation (e.g., heatmaps in Fig. 8–9) without quantitative metrics for concept interpretability (e.g., concept-accuracy scores via human trials or downstream tasks).
- The MCC metric measures alignment but does not assess disentanglement quality (e.g., mutual information between recovered concepts).

**Other Comments Or Suggestions:**

1. Include a discussion on how structural diversity could be measured/validated in practice (e.g., via concept-class mutual information).
2. Compare with concurrent work on concept bottlenecks (Delfosse et al., 2024) to highlight differences in identifiability guarantees.
3. Release code for reproducibility, especially for Jacobian regularization and synthetic data generation.

**Other Strengths And Weaknesses:**

see above.

**Questions For Authors:**

The questions are listed above.

**Relation To Broader Scientific Literature:**

The work extends identifiability theory in concept learning, contrasting with parametric approaches (e.g., linear ICA [Hyvarinen & Morioka, 2016], additive models [Lachapelle et al., 2023]). It aligns with cognitive-inspired methods [Brady et al., 2023] but diverges by eliminating parametric constraints. The structural diversity condition parallels sparsity assumptions in nonlinear ICA [Zheng et al., 2022], though the focus on class-concept relationships is novel.

**Theoretical Claims:**

I checked the correctness of proofs. All proofs are correct.

---

> ### Author Rebuttal · Authors · 2025-04-01
>
> We are profoundly thankful for your insightful feedback. In light of it, we have included new discussions and conducted **new experiments** (https://anonymous.4open.science/r/0-518C/rebuttal_new_results.pdf). Please find our point-by-point responses below:
>
> **Q1:** Thm. 1 in high-dimensional space or under noisy observations.
>
> **A1:** Thank you for the question. Theoretically, the assumption only requires Jacobians to be linearly independent at some points to ensure meaningful variation—which can be expected from large populations. We have also added new experiments in light of your suggestions:
>
> - **High-dimensional space:** Beyond the five image datasets used before, we added two more (Figs. 5–8, Table 1 in the link). Since images are naturally high-dimensional, these results across seven datasets support applicability in such settings.
>
> - **Noisy observations:** We added experiments with noisy images (Figs. 3-4 in the link), showing meaningful concepts remain identifiable.
>
> **Q2:** Practical applicability of structural diversity.
>
> **A2:** Thanks for the suggestion. We have added new experiments under partial violations (Figs. 1–2 in the link), showing robustness. Intuitively, if a concept (e.g., “furry”) is absent in some classes, it can be disentangled. If all classes share it, it is unidentifiable under Thm. 2. However, our local comparison framework (Thm. 1, Prop. 1) still ensures identifiability even without global structural diversity.
>
>
> **Q3:** Reproducibility details for Jacobian regularization and synthetic data.
>
> **A3:** In light of your suggestion, we have highlighted details further in the updated version. Due to the rebuttal policy, we can only include figures in the anonymous link. We approximate ℓ₀ regularization with an ℓ₁ norm (L369 right) since ℓ₀ norm is known to be indifferentiable. We use a flow-based estimator (Sorrenson et al., 2020) to recover latent concepts (L365–367 right), though our theory is estimator-agnostic—as long as the observed distribution is matched and regularization applied. For synthetic data, we randomly construct a class-concept structure satisfying our conditions and sample each latent variable conditioned on its class labels.
>
>
> **Q4:** Suggestions on empirical evaluations.
>
> **A4:** Thank you so much. We added the following new experiments accordingly:
>
> - **Q4.1:** Partial violation of structural diversity.
>
>    **A4.1:** Evaluated in Figs. 1–2 in the anonymous link; results show robustness.
>
> - **Q4.2:** Quantitative metric for real-world evaluations.
>
>    **A4.2:** We added Fréchet Inception Distance (FID) and Perceptual Path Length (PPL). As shown in Table 1 in the link, ours consistently outperforms baselines.
>
> - **Q4.3:** More metrics for simulation.
>
>    **A4.3:** We added DCI score and Mutual Information Gap (MIG) (Figs. 1–2 in the link); same conclusion holds.
>
> **Q5:** Entangled concepts on FFHQ.
>
> **A5:** Thanks for your question. Learning by local comparisons focuses on the unique part of each class (e.g., a group of dependent concepts), enabling partial identification even when full disentanglement is impossible. FFHQ illustrates this: despite global violations, concept groups remain identifiable (L1460–1467). In addition to the full identifiability (shown in other datasets), this partial identifiability is also one of our main focuses.
>
> **Q6:** Baseline comparisons lack details.
>
> **A6:** Thanks. As noted in L407 left, all models share the same setting except the data-generating process. We have highlighted this further in the updated manuscript.
>
>
> **Q7:** Some steps (e.g., Eqs. 82–83) lack intuition.
>
> **A7:** Thank you—we have added more intuition accordingly. For any $i \in \\{1,\ldots,n_A\\}$ and $v \in \\{1,\ldots,n_A\\} \setminus i $, $(\pi(i), v) \notin \mathcal{T}$ (apologies for a typo—$k$ in L1053 should be $v$) implies that all entries in row $\pi(i)$ of matrix $\mathrm{T}$ are zero except possibly at index $i$. Since $\mathrm{T}$ is inveritble, each row must have a non-zero entry, implying $(\pi(i), i) \in \mathcal{T}$.
>
>
> **Q8:** Measuring structural diversity in practice.
>
> **A8:** Thanks for the excellent point. We have added a discussion in the updated version. Structural diversity can be assessed via concept–class mutual information (as you kindly suggested) and domain knowledge (e.g., “sunshine” may be universal, “furry” is class-specific). Even if violated or unsure, our local comparison framework ensures alternative identifiability.
>
> **Q9:** Compare with Delfosse et al., 2024 on differences in identifiability guarantees.
>
> **A9:** Thanks for the suggestion. For Delfosse et al. (2024), did you mean *“Interpretable concept bottlenecks to align reinforcement learning agents”*? We are not sure if we understood correctly, as we did not find identifiability theories in that paper. Please let us know if we missed or misunderstood anything—we would be more than happy to include a discussion accordingly.

---

### Official Review · Reviewer_DXpa · 2025-03-13

**Overall Recommendation:** 4

**Summary:**

The authors tackle the important problem of identifying latent classes from data. They claim that given some constraints, it should be possible to do given that different classes (e.g. different animals), will produce different sets of observable concepts (i.e. different shapes, colors, etc); and that as long as there is at least one difference, it should be possible to identify them. The formalize these notions via several theorems and definitions. Finally, they validate these empirically by fitting simple non-parametric matrices corresponding to each pair of class and related concepts. Specifically, they fit a sparse model that attempts to fit concepts that are generated according to their theoretical assumptions by each class. The compare these to other data generating mechanisms that are entangled or do not have sparse structure and show they are harder to fit.

**Claims And Evidence:**

Yes. The authors theoretically define their model and compare it empirically to one that does not follow their regularization schemes.

**Essential References Not Discussed:**

Not that I can think of.

**Experimental Designs Or Analyses:**

For the most part yes. The simple toy experiments are easy to follow. I have questions about the real world datasets. Specifically, what models did the authors use? I could not find this.

**Methods And Evaluation Criteria:**

Yes. The authors extensively test their theoretical approach on several datasets.

**Other Comments Or Suggestions:**

I suggest the authors re-write the paragraph in lines 82-92. If two classes of observations share an identical set of concepts are they really two different classes? It seems like the very set of concepts in the observation is what defines a class (from the perspective of the AI/Human) and not the other way around. To put it another way, if the observations are really generated by different classes, eventually they should differ, if they don't they are effectively the same (assuming we can observe the differences). Also, add a reference from the cogsci literature to the actual mechanisms you are taking inspiration from.

I really like that the authors explain the intuitions behind their theorems and definitions. My only suggestion would be to introduce these intuitions first and the theorems later. A general structure could be: "Intuitively, structural diversity is important ...." and then "We can formalize this notion through theorem ....". I feel like this is a more natural way to read things than just suddenly having to wade through formulas.

**Other Strengths And Weaknesses:**

The main strength of the paper is rigorous the authors try to be. I don't know if all the theorems are correct though. Main weakness is that for the real world datasets it is not clear how the concepts related to each class are extracted. A common story I see in these articles is that they abstract away the process of extracting the relevant entities (in this case the class-related concepts) and operate directly on those. If that is what's happening here then that makes the whole article much less useful since part of what makes this task hard is that we need to extract these concepts from raw input.

**Questions For Authors:**

Can you provide more details about the models used to test on real world data?

**Relation To Broader Scientific Literature:**

I think the authors do a good job to refer to the related literature. I feel like some of their theoretical claims are very similar to previous work, especially in the ICA literature, but they are very thorough about formalizing their ideas.

**Theoretical Claims:**

I didn't check all the theorems. Some are intuitive, while I admit that others are a bit hard to follow given the notation. For example, theorem 2 and the corresponding figure (Figure 4), were not completely clear to me.

---

> ### Author Rebuttal · Authors · 2025-04-01
>
> We sincerely appreciate your constructive feedback. Accordingly, we have included several new discussions in the updated version. Please see our point-by-point responses below:
>
> **Q1:** More intuition on Theorem 2 and Figure 4.
>
> **A1:** Thanks for the suggestion. We have added more discussion in the revised version. Theorem 2 guarantees the identifiability of each class-dependent concept under the structural diversity condition. Figure 4 illustrates this condition: for example, for concept $\mathbf{z}_1$, there exist classes $\mathbf{c}_1$ and $\mathbf{c}_3$ s.t. $\mathbf{z}_1$ is unique to $\mathbf{c}_1$, allowing that concept to be disentangled. Meanwhile, class-independent concepts can also be disentangled as a block by leveraging distributional variety.
>
> **Q2:** What models did the authors use for the real-world datasets?
>
> **A2:** Thanks for the question. We used a general incompressible-flow network (GIN) following Sorrenson et al., 2020, with an additional sparsity regularization (L365-370 right). We have further highlighted this in the updated manuscript according to your constructive comments.
>
> **Q3:** Relation of theoretical claims with previous work, especially in the ICA literature.
>
> **A3:** Thank you for the insightful comment. We truly appreciate the opportunity to further illustrate how our theoretical results relate to prior work. While Thm. 2 shares ICA’s goal of identifying latent variables, it differs in assumptions, techniques, and scope. We would be very grateful for any further feedback if any part of the following differences is unclear or could be better articulated.
>
> - ***Objective:*** While ICA aims to recover all latent variables individually, our focus is on identifying which concepts are reliably recoverable, grounded in the cognitive process of comparison. Our main contribution lies in learning by local comparisons—showing that unique concepts can be disentangled from any class pair. This naturally extends to global identifiability (Thm. 2), but the local view enables more flexible, practical analysis. In contrast, most previous work in ICA cannot provide any guarantee with partial violation of their assumptions for some latent variables.
>
> - ***Techniques:*** ICA methods are built on elegant and powerful assumptions—such as auxiliary variables or sparse mixing—that enable strong identifiability results. For example, iVAE needs more than 2n+1 environments with sufficiently varying distributions; sparsity-based methods assume sparse concept–observed mixing structure. Our method avoids these by relying on class–concept structures, offering partial identifiability when global assumptions may not fully apply.
>
> In summary, our theory targets a different question: determining which concepts are identifiable under minimal, localized assumptions. We do not claim generality over ICA, but rather provide complementary insights, especially in settings where global assumptions may be partially violated.
>
> **Q4:** How to extract the concepts from raw input?
>
> **A4:** Thank you for the great question, and we have added more details in the updated version. After estimation (detailed in A2), we recover a latent vector from observed data and class labels. Each dimension of this vector corresponds to a concept. To aid interpretation, we rank dimensions by their variances—based on the heuristic that higher-variance components are often more semantically meaningful.
>
> **Q5:** Rewrite the paragraph in lines 82-92, and add a reference from the cogsci literature.
>
> **A5:** Thanks so much for the great suggestion. We have rewritten the paragraph as follows and added a line of references.
>
> > We address this question by grounding our approach in a fundamental cognitive mechanism: humans learn concepts by contrasting diverse classes of observations. Classic studies have shown that concept formation relies on detecting distinctions across examples—Bruner et al. (1956) emphasized learning through contrasts between exemplars and non-exemplars; Gibson (1963, 1969) proposed differentiation as a basis of perceptual learning in infants; and Gentner & Namy (1999) demonstrated that direct comparison enables children to abstract category-defining features. This principle is further supported by extensive literature across cognitive science, reinforcing that learning by comparison is the key underlying engine.
>
> ---
>
> References:
>
> *Bruner, J. S., Goodnow, J. J., & Austin, G. A. (1956). A Study of Thinking. Wiley.*
>
> *Gibson, E. J. (1963). Perceptual learning. Annual Review of Psychology.*
>
> *Gibson, E. J. (1969). Principles of perceptual learning and development. Appleton-Century-Crofts.*
>
> *Gentner, D., & Namy, L. L. (1999). Comparison in the development of categories. Cognitive Development.*
>
> ---
>
> **Q6:** Move intuitions before the theorems.
>
> **A6:** Thank you very much for the helpful suggestion. We have revised the layout accordingly, placing the intuitions before the theorems for better flow.

---

### Decision · Program_Chairs · 2025-05-01

**Decision:**

Accept (poster)

**Comment:**

The reviewers unanimously acknowledge the quality of the contribution and recommend acceptance. I concur that this is a solid contribution to the program and recommend acceptance.